

# DOM quality in a peatland and forest headwater stream: seasonal and event characteristics

Tanja Broder[1, 2], Klaus-Holger Knorr[2] Harald Biester[1]

[1]IGÖ, Umweltgeochemie, TU Braunschweig, Langer Kamp 19c, 38106 Braunschweig, Germany
[2]ILÖK, Hydrologie, WWU Münster, Heisenbergstr. 2, 48149 Münster, Germany

*Correspondence to*: Tanja Broder (broder@uni-muenster.de)

**Abstract.** Peatlands and peaty riparian zones are major sources of DOM, but are poorly understood in terms of export dynamics and controls thereof. The quality of DOM affects function and behavior of DOM in aquatic ecosystems, but DOM quality can also help to track DOM sources and their export dynamics under specific hydrologic preconditions. The objective of this study was to elucidate controls on temporal variability in DOM concentration and quality in discharge from a bog and a forested peaty riparian zone, particularly considering drought and storm flow events. DOM quality was monitored using spectrofluorometric indices for aromaticity ($SUVA_{254}$), apparent molecular size ($S_R$) and precursor organic material (FI), as well as PARAFAC modeling of excitation emission matrices (EEMs).

Indices for DOM quality exhibited major changes due to different hydrologic conditions, but patterns were also dependent on season. The forested site with mineral, peaty soils generally exhibited higher variability in concentrations and quality compared to the outflow of an ombrotrophic bog. During snowmelt and spring events surface-near protein-like DOM pools were exported. During drought a microbial DOM fraction originating from groundwater and deep peat layers was exported, characterized by high FI, high $S_R$ and increasing humic-like fluorophores C1% or C4%. During discharge events this deep sourced DOM pool was diluted by humic-rich surface-near DOM pools, which were more aromatic, also of microbial origin with high FI and $S_R$, and an increase in humic-like fluorophores C2% and C3% at the forested site. The FI suggested export of DOM of strong microbial alteration particularly under discharge events with dry preconditions. This might be due to accelerated microbial activity in the peaty riparian zone of the forested site. Our study demonstrated that DOM export dynamics are not only a passive mixing of different hydrological sources, but assessing DOM quality can greatly improve our understanding of DOM sources and their response under different hydrological preconditions.

Keywords: DOM, PARAFAC, Peatlands, riparian zone, forest soils, fluorescence

## 1 Introduction

Dissolved organic matter (DOM) is ubiquitous in soils and aqueous ecosystems. It plays a fundamental role in surface water chemistry, e. g. in metal bioavailability and mobility (Tipping et al., 2002), nutrient cycling (Jansson et al., 2012), pH



buffering and ionic balance (Hruška et al., 2003). It affects light penetration (Karlsson et al., 2009), the aquatic food web structure (Jansson et al., 2007), is an energy source for microbial metabolism (Cole et al., 2007; Amon and Benner, 1996) and finally part of the carbon cycle (Cole et al., 2007). But not only DOM quantity is of great interest, as the DOM quality strongly affects function and behavior of DOM in aquatic ecosystems.

Main DOM input to aquatic systems is of terrestrial origin. Concentrations and characteristics of DOM vary strongly among surface waters depending on catchment, climate and hydrology (Ågren et al., 2014; Laudon et al., 2004; Frost et al., 2006; Winterdahl et al., 2014). However, DOM concentrations and characteristics can also vary largely over time due to seasonal changes in production, consumption and transport of DOM. Peatlands, which store large amounts of carbon have thereby received attention as a major source of DOM to surface water (Worrall et al., 2002; Aitkenhead et al., 1999). But also wet

riparian zones with organic-rich layers are recognized as DOM source (Bishop et al., 2004; Seibert et al., 2009; Laudon et al., 2004). The annual dissolved organic carbon (DOC) concentration dynamics and long-term DOC concentration increase observed for many catchments (Monteith et al., 2007; Worrall et al., 2004) points out the importance to understand DOM origin and factors controlling DOM export. Storm events have been shown to be quantitatively important for DOM exports to streams in peatland catchments (Clark et al., 2007) and carbon-rich riparian zones, as they generate high DOM

concentration peaks. Especially for drinking water production not only DOM quantity, but also DOM quality is important. Especially aromatic structures of DOM could cause disinfection-by-product (DBPs) generation during drinking water treatment (Korshin et al., 1997). More aromatic, humic DOM also decreases light penetration and DOM photo-degradation potential in surface waters (Cory et al., 2007, Ward and Cory, 2016). The easily biodegradable DOM (BDOM) fraction - mainly fresh, protein-like DOM, derived from root or leaf exudates, litter decay or leachates - can be readily utilized and

serves as important nitrogen and phosphorus source in aquatic systems (Fellman et al., 2009). Microbially processed DOM are residual and recalcitrant substances. Assessing such variability in DOM quality can be a valuable tool to track DOM sources and transport mechanisms (Singh et al., 2014), which is crucial for predicting DOM exports and quality.
Comprehensive studies mainly focused on total DOM concentration and much is known about the DOM export from peatland and forested catchments (e. g. Laudon et al., 2011; Grabs et al., 2012; Clark et al., 2009). Trends of DOM quality

during stormflow events or differences depending on catchment type are scarce. (Inamdar et al., 2011) and (Hood et al., 2006) characterized DOM during storm events in a temperate forest catchment dominated by mineral soils. A high contribution of aromatic structures during storm flow was ascribed to flushing of humic-rich near-surface soil layers and lower contribution of shallow groundwater. Organic soil layer DOM can be highly aromatic or humic, reflecting decomposition of complex plant and soil organic matter. As DOM percolates through the soil sorption to mineral phases

preferentially removes larger, aromatic components (e.g. Meier et al., 2004; Kaiser and Zech, 2000) and longer residence times enhance alteration of DOM by microbial processes. Thus groundwater DOM is mostly of microbial origin and of apparently smaller molecular size. While (Singh et al., 2014) described a strong pulse of protein-like DOM during autumn leaf fall, (Perdrial et al., 2014) perceived only modest shifts in DOM quality over seasons in a forested catchment in New Mexico. (Fellman et al., 2009) focused on the bioavailable fraction of DOM from wetland and forest soils, and found a



strong biotic control on BDOM interacting with abiotic processes and hydrologic flowpaths. The BDOM fraction was highest during spring due to a low biotic demand and shallow flow paths. (Ågren et al., 2008) reported higher aromaticity and apparent molecular size during snowmelt at a wetland catchment compared to a forest catchment. However, a comparison to other high discharge events during the growing season is lacking.

A limitation in DOM quality studies is that determination of DOM structures is elaborate and expensive, while large datasets and high temporal resolution would be desirable (Strohmeier et al., 2013). UV-Vis and fluorescence spectroscopy are limited in data interpretation in terms of specific chemical structures, but due to low cost and rapid analysis enables us to generate a comprehensive dataset covering a wide range of hydrologic and seasonal conditions. It allows distinguishing between different DOM constituents and disentangling their specific export behavior and might also be used to trace different DOM

sources (Hood et al., 2006). Several optical indices describe the nature of DOM: Specific Ultra-Violet Absorbance at 254 nm ($SUVA_{254}$) is commonly used as indicator for the proportion of aromatic structures (Weishaar et al., 2003). The spectral slope ratio ($S_R$) (Helms et al., 2008) is used as a proxy for apparent DOM molecular size. The humification index (HIX) (Ohno, 2002) and fluorescence index (FI) (Cory and McKnight, 2005) are derived from fluorescence-based excitation-emission matrices (EEMs). While HIX describes the degree of humification, FI differentiates between plant derived and

microbial or planktonic derived DOM. The fluorescence EEMs can be further analyzed using parallel factor analysis (PARAFAC), decomposing the EEMs into hypothetic fluorophores related to differences in composition of DOM (Stedmon and Bro, 2008; Murphy et al., 2013).

Following up on a previous study describing DOC fluxes and concentration dynamics from a bog catchment (Broder and Biester, 2015), the present study is intended to elucidate changes in DOM quality and dynamics comparing a bog and a

forested peaty riparian zone, as those landscape types are considered as the main sources of stream DOM in the catchment. These dynamics are expected to be different as the two sub-catchments differ in vegetation, hydrology and soil properties. We expect water level, temperature, and precipitation to be the main controls on stream DOM concentration and quality. On the one hand changes in DOM will depend on season, which is attributed to DOM production and consumption. On the other hand hydrological connectivity of different source areas will also affect stream DOM composition, because shallow

groundwater or deeper peat layer versus organic rich upper soil layer or surface-near peat layer, can differ in DOM composition. The effects of drought and storm events were particularly considered, as they induce major DOM dynamics due to changes in hydrologic flow paths in the catchment, such as a development of surface flow networks or a connection of organic rich surface layers to discharging waters. The characterization of stream DOM is used to track DOM sources and their dynamics under specific hydrologic (pre)conditions to elucidate the role of this specific landscape areas in regulating

the stream DOM dynamics.

For this purpose we chose a headwater stream catchment to compare DOM export from discrete landscape units. Short residence times make in-stream processes negligible and enables landscape-type specific studies. For DOM characterization we used spectrofluorometric indices like $SUVA_{254}$, $S_R$, HIX and FI, as well as PARAFAC modeling.



## 2 Materials and Methods

### 1.1 Study site

The study site is located within the nature protection area of the Harz Mountains (Fig. 1). The Odersprung bog exhibits an erosion rill, draining the peatland. The catchment responds quickly to rainfall events and discharge is mainly fed by surface-near waters. A more detailed hydrologic description is given in (Broder and Biester, 2015). The bog vegetation is dominated by *Sphagnum magellanicum* and *S. rubellum*, associated with *Eriophorum angustifolium* and *Molina caerulae* (Baumann, 2009). The peatland is surrounded by spruce forest growing on a cambic podzol soil at the hillslopes and peaty soils with deep organic topsoil layer in the riparian zone. One discharge sampling was conducted directly at the rill outflow, where all water originates exclusively from the bog. Another sampling spot was established about 20 m further downstream where the small headwater stream increasingly receives water from the surrounding forested, organic rich mineral soils and peaty riparian zone. The catchment is underlain by granitic bedrock. Bog mean peat thickness is about 3 m, while the mineral soils are shallow at the hillslope (30 cm) and deeper in depressions (100 cm). Organic content of the soil varies between 30 % and 97 % in the organic-rich surface layers (Broder & Biester, unpublished data).

### 1.2 Sampling and field measurements

Discharge sampling at each sampling spot was conducted from snowmelt to begin of snowfall in 2013. Water samples of 500 mL volume were taken by an automated water sampler (Teledyne ISCO, USA) in a six day interval. Additional grab samples were taken every two to three weeks in PE tubes, which were previously rinsed twice with sample. High frequency storm event sampling was conducted on several occasions in a 3 hour interval. A V-notch weir was installed at the bog outlet for discharge quantification. Water stage at the weir as well as at the bog site was recorded by a water level logger (Odyssey dataflow systems, New Zealand) installed in a slotted PVC piezometer tube of 4 cm diameter. Temperature, humidity and precipitation was monitored on site (using a tipping-bucket rain-gauge and tinytag tgp 4500 and 4810, Gemini, Belgium).

### 1.3 Laboratory Analysis, indices and PARAFAC modelling

Water samples were vacuum filtered with a 0.45µm nylon filter (Merck Millipore, Germany) and stored in the dark at 4°C. All water samples were analyzed for DOC by thermo-catalytic oxidation using the NPOC method (non-purgeable organic carbon; multi N/C 2100S, Analytik Jena, Germany). UV-VIS spectra of all samples were recorded on a Lambda 25 (Perkin Elmer, USA) in a range of 200-800 nm at a 0.5 nm resolution. For subsequent fluorescence spectroscopy, samples were diluted to absorption <0.3 at 254 nm to reduce inner filter effects. Absorbance at 254 nm wavelength ($abs_{254nm}$) was used as indicator for the absolute aromaticity of DOM samples as conjugated systems like aromatic molecules have the greatest absorption in the near UV range. $SUVA_{254}$ was calculated by dividing absorbance at 254 nm (in $m^{-1}$) by the DOC concentration (in $mg\ L^{-1}$) according to (Weishaar et al., 2003) with increasing $SUVA_{254}$ values indicating a higher aromaticity. The spectral slope ratio ($S_R$), a proxy inversely related to molecular weight, was calculated after (Helms et al.,



2008) by dividing the slope in the interval of 275-295 nm by the slope at 350-400 nm. Slopes were determined using linear regression of log-transformed absorption spectra.

Fluorescence excitation-emission matrices (EEMs) were collected with a Cary eclipse fluorescence spectrometer (Agilent, USA) in 5 nm steps over an excitation range of 240-450 nm and 2 nm steps over an emission range of 300-600 nm. Inner

filter correction, blank subtraction and Raman normalization was performed using the drEEM 0.2.0 toolbox from (Murphy et al., 2013) and MATLAB (Version 2013a, MathWorks, USA). Reshaped EEMs were subjected to parallel factor analysis (PARAFAC) to obtain hypothetical fluorophores for DOM fingerprinting. In total, 435 samples were included in the PARAFAC model, both discharge and pore water samples originating from different sites. Samples examined in this study accounted for 242 samples within this model. A five fluorescence component model could be obtained and split-half

validated following the drEEM and N-way toolbox (Murphy et al., 2013 and Stedmon and Bro, 2008). The sum of fluorescence intensities of the modeled components thereby represents the total fluorescence of a sample. The contribution of fluorescent DOM (fDOM) to total DOC was evaluated by normalizing total fluorescence by DOC concentrations (fDOM/DOC ratio).

The FI was calculated by the ratio of fluorescence emission intensities at 470 nm and 520 nm at an excitation wavelength of

370 nm ((Cory and McKnight, 2005). The FI differentiates between vascular plants derived (FI: 1.3 - 1.4) and microbial or planktonic derived DOM (FI: 1.7 – 2.0) (McKnight et al., 2001) as the ratio represents the greater decrease in emission with increasing wavelengths of microbial derived DOM. As our study site is a headwater catchment we assume that all DOM is of terrestrial origin and therefore, we interpret an FI > 1.7 as microbial derived or microbial processed DOM. The humification index (HIX) was calculated after the modified equation of (Ohno, 2002) whereby higher values in a range of 0 to 1 indicate a

red-shift of spectral emission and a higher degree of DOM humification.

# 3 Results

## 3.1 Seasonal trends

### 3.1.1 Hydrologic conditions and DOC concentrations

The DOC concentration record and hydrologic characteristic at the bog site has been described previously in (Broder and

Biester, 2015). In short, bog discharge exhibited a flashy regime with an instantaneous response to rain events. The rain event with the highest recorded discharge peak occurred in spring, while in summer a longer drought period resulted in very low discharge and little response to rain fall due to recovery of water storage within the bog. More frequent rain events in fall at wetter antecedent moisture conditions caused again more flashy discharge and concentration responses.

The variability of DOC concentrations over the year was greater than during single rain events at both sites ranging between

5 to 45 mg L$^{-1}$ (Fig. 2). However, rain events were responsible for the highest recorded DOC concentrations during a sampled rain event in fall. The concentration trend generally followed the vegetation period at both sites with highest





concentrations in late summer and fall. Lowest concentrations were measured during snowmelt at both sites. Concentrations of DOC were higher at the forested site than at the bog outlet over the whole sampling period.

### 3.1.2 DOM quality using spectrofluorometric indices

The $abs_{254nm}$ as index for total aromaticity of the DOM exhibited a similar trend as the DOC concentrations at both sites over the year (Fig. 2). Nonetheless, $SUVA_{254}$ values as index for proportional aromaticity of DOM, varied between 3.5 and 5.3 at the bog site and between 3.5 and 6 at the forested site, but with no seasonal trend as observed for DOC concentrations. According to (Weishaar et al., 2003) calculated $SUVA_{254}$ values corresponded to a DOM aromaticity of 27 – 38 % for the bog site and 29 - 42 % for the forested site. Variations were mainly induced by hydrologic conditions with high values during rain events in spring (up to 5.5). Mean $SUVA_{254}$ values were higher at the forested site and showed a larger variability than at the bog site. During the summer drought period $SUVA_{254}$ values decreased at both sites, but stronger at the forested site.

As expected, the $S_R$ as reciprocally proportional index for molecular weight of DOM exhibited a rather opposite trend than $SUVA_{254}$ (Fig. 2). The annual dynamic was similar at both sites, but with higher $S_R$ during snowmelt ($S_R$ up to 2.2) and sampled rain events ($S_R$ 1.7-2.0) at the forested site, indicating a lower molecular weight. During the summer drought $S_R$ steadily increased from 1.7 to 2.0 indicating decreasing molecular weight. With the onset of fall rain events molecular weight increased again indicated by lower $S_R$ values.

The humification index (HIX), as well as the fluorescence index (FI) also exhibited no annual trend (Fig. 3). At the bog and the forested site HIX only varied during snowmelt and spring events with lower values down to 0.75 and 0.8, respectively, compared to HIX in summer and fall, where values remained between 0.90 - 0.94 and 0.91 – 0.95, respectively. FI exhibited similar values at the bog and forested site between 1.5 and 1.75. At the forested site FI increased during summer drought (from 1.5 to 1.7).

### 3.1.3 Hypothetical fluorophores modeled by PARAFAC

PARAFAC modeling resulted in a five component model with four humic-like and one protein-like hypothetic fluorophores. Excitation-Emission regions of each component can be found in Fig. 4. The modeled PARAFAC component C1 can be compared to a terrestrial, humic-like fluorophore originating from forests and wetlands soils, as described by (Perdrial et al., 2014) and C2 can again be described as humic-like (see e.g. C3 of Singh et al., 2014), but the excitation-emission region is shifted to higher excitation and emission wavelengths compared to C1 indicating more conjugated and more aromatic fluorescent molecules. A component similar to C3 has previously been described as humic, but also of terrestrial origin, small molecular size, recalcitrant and reduced (Cory and McKnight, 2005; Singh et al., 2014; Fellman et al., 2008; Perdrial et al., 2014). C4 is only slightly shifted compared to the excitation-emission region of C3 and compares to C2 from (Fellman et al., 2008) and (Ohno and Bro, 2006), another humic-like fluorophore. C5 could be described as tryptophan-like, of microbial origin, labile and of recent biological production (described in e. g. (Fellman et al., 2008) (C8)). It can be used as





proxy for BDOM (Fellman et al., 2008). In C1 and C3 also fulvic-like fluorophores might be included, which are more hydrophilic and therefore more mobile than the humic-like DOM (Fellman et al., 2008 C3, C4), but could not be clearly separated in individual components.

### 3.1.4 DOM quality using PARAFAC

The fDOM, as the sum of all fluorescent components modeled by PARAFAC, showed changes with discharge events with both minimum and maximum intensities during sampled discharge events (Fig. 5). Fluorescence was elevated during summer and lower values occurred in spring and fall at both sites. Normalizing fDOM to DOC concentrations, i.e. monitoring changes in fDOM composition in percentage of each component, a decrease in the fluorophore fraction in DOM from spring to fall could be observed (Fig. 6). This fluorophore fraction decreased also during individual sampled rain events.

The fDOM at the bog site showed little seasonal changes in the contribution of the four components over the year, only C4% exhibited a similar seasonal trend as DOC concentrations. Greatest changes were perceived during snowmelt with highest protein-like C5% (~10-17%) and variable humic-like C1%, C2% and C4% contributions at the bog site. The protein-like C5% contributed least to fDOM with about 5% during most of the record. The components C1% and C4% increased during summer drought at the bog site.

The forested site showed a greater variability in component contributions over the entire record, but no seasonal trend. The protein-like C5% contributed largely to fDOM during snowmelt and spring events. The humic like C1% increased during summer drought, while C2% deceased at the same time. The components C2% and C3% were elevated at the forested site, while higher contributions of C4% were more attributed to the bog site (Fig. 6). Overall, the C4% can be denoted as a bog derived humic-like component, while C2% and C3% were predominantly forest derived humic-like components. The humic-like C1% and the protein-like C5% could not be specifically attributed to one of the sites.

### 3.2 Snowmelt, drought and rain events

### 3.2.1 Snowmelt

Snowmelt samples exhibited a distinct pattern compared to discharge events in other seasons. DOC concentrations (5 - 12 mg L$^{-1}$), abs$_{254nm}$ (40 - 140 m$^{-1}$), HIX (0.75 - 0.9), and fDOM (0.6 - 1.1) were lowest and normalized fDOM/DOC (0.06 – 0.14) was highest during snowmelt at both sites. Values of S$_R$ during snowmelt were high at the forested site with about 1.95 - 2.2, while S$_R$ at the bog site exhibited mean values of 1.8 - 1.9 (Fig. 7), which were still higher than following spring samples, though. SUVA$_{254}$ was lower (4 – 4.5) than in following spring at the forested site, while at the bog site SUVA$_{254}$ of 4 – 4.5 were similar to values of the rest of the record. The protein-like PARAFAC component C5 % was elevated during snowmelt and contributed 8 – 16 % to total fluorescence at both sites (Fig. 8). Values of the humic-like C2% and C4% were low at the bog site, while at the forested site the bog derived C4% was almost absent and the humic-like forest-derived C3%




was strongly elevated (contributing about 20 % to total fluorescence, Fig. 8). It should be noted though, that the snowmelt event was not entirely covered by our sampling period as the upstream water was still covered with snow.

### 3.2.2 Spring

The sampled spring rain events could be differentiated by hydrologic preconditions as the first event occurred under dry preconditions, while the second described spring event followed after preceding rain events under wet hydrologic preconditions. DOC concentrations and $abs_{254nm}$ were high after dry preconditions (21 - 33 mg $L^{-1}$ and 200 – 360 $m^{-1}$) and much lower after wet preconditions (8 - 24 mg $L^{-1}$ and 80 - 300 $m^{-1}$) at both sites (Fig. 7). $SUVA_{254}$ values at the forested site were persistently high during the whole spring time (4.5 - 5.5), but decreased during each rain events. At the bog site $SUVA_{254}$ values were also decreasing during spring rain events, but kept similar values throughout the rest of the year (3.6 – 4.6). $S_R$ exhibited generally higher values during both spring events indicating smaller DOM molecular size at the forested site. High resolution sampling before the wet preconditions spring event during low flow conditions showed a steady decrease in $S_R$ until the onset of the rain event at the forested site. However, at the first spring event with dry precondition the $S_R$ values peaked only at the declining limb of the hydrograph. At the bog site $S_R$ values during spring events were low, but again with increasing values at the declining limb of the hydrograph, which indicates a decrease in apparent molecular size.

Contribution of protein-like component C5% exhibited elevated values during spring events at both sites and higher values at the second event with wet preconditions at the bog site (Fig. 8). The bog derived humic-like C4% was at both sites at both events decreasing. This trend was more distinct at the forested site with a quick drop to zero at the second event. The humic-like C1% dropped from the first sampled spring event to the second one at both sites, while the more aromatic C2% increased at the forested site from the first to second event. The predominantly forest derived humic-like C3% also increased at all storm events.

### 3.2.3 Drought

The year 2013 was characterized by a strong summer drought, which caused low discharge over the summer months. During the prevailing drought period DOC concentrations, $abs_{254nm}$ and $S_R$ increased, while $SUVA_{254}$ decreased especially at the forested site (from 5.4 to 3.7, Fig. 2). The humic-like C1% increased and the humic-like, but more aromatic C2% decreased at the forested site, while at the bog outlet an increase in the humic-like bog-derived C4% was perceived (Fig.6).

### 3.2.4 Fall

The fall event following the summer drought generated the highest DOC concentrations of the annual record with 45 mg $L^{-1}$ DOC at the forested site and 40 mg $L^{-1}$ DOC at the bog site. Even though, $abs_{254nm}$ was high and even increasing during the event at the forested site, $SUVA_{254}$ values were lower than during spring events indicating a lower aromaticity of DOM in fall (Fig. 7). Congruently, $S_R$ values were higher at both sites, indicating smaller DOM during fall event. The humic-like





PARAFAC components C1%, C4% and the protein-like C5% were decreasing during the fall rain event at the forested site, while the forest derived humic-like C2% and C3% increased (Fig. 8). The bog derived C4% contributions were higher than in spring, but with the same decreasing trend with the ongoing rain event at both sites. The protein-like C5% exhibited lower values during the fall event than at the spring rain events with slightly decreasing values at both sites.

**4 Discussion**

A comparison of the different spectrofluorometric indices confirmed their general suitability to track origin and dynamics of DOM. As expected, the spectroscopic indices $SUVA_{254}$ and $S_R$ were inversely related. This implies that an increase in aromaticity also caused an increase in apparent molecular size and aromatic structures seem to dominate the high molecular size DOM fraction in this study. Fluorometric indices exhibited different trends than UV-VIS indices even though similar

DOM qualities should be predicted. HIX as fluorometric index for humic components showed less variability than $SUVA_{254}$. A difference between those two indices was also reported by (Inamdar et al., 2011). As $SUVA_{254}$ is a proxy exclusively for aromatic DOM, HIX might include other humic or hydrophobic DOM components derived from plant decomposition. Trends of HIX in this study rather indicate that HIX represents all fluorescent humic-like components, which resulted in less variability due to a domination of humic components in all catchment compartments.

The FI values of both sites varied between 1.5 and 1.75. This indicates a domination of microbial or microbial processed DOM over vascular plants derived DOM in the catchment (Cory and McKnight, 2005). As we assume that all DOM is of terrestrial origin it implies that the majority of measured stream DOM is not fresh organic material as would be indicated by high contributions of C5%, nor highly aromatic DOM derived from plant decomposition, but has been strongly modified by microbial processes within the soils. It also indicates long residence times of DOM in the soils, as produced DOM is mainly

not immediately transported to surface waters, but gets altered or consumed within the soils of the catchment. This may be supported by the high $SUVA_{254}$ values which indicate a strong aromatic fraction contributing 27 to 42% to DOM, presumably due to residual enrichment.

In contrast to other studies, the forested site exhibited higher $SUVA_{254}$ values most of the time. For example, (Ågren et al., 2008) and (Wallin et al., 2015) found similar $SUVA_{254}$ values or even higher values at a peatland site, respectively,

compared to mineral soils at a forested site. On the one hand this might be due to the dominating *Sphagnum* vegetation and peat at the bog site here, as *Sphagnum* is known to produce less aromatic organic matter than vascular plants, which also occur in peatlands, due to the lack of lignin in *Sphagnum* mosses (Spencer et al., 2008). On the other hand, the domination of peaty soils in the riparian zone and the domination of shallow sub-surface flow over groundwater contribution at the forested site might enhance DOM aromaticity due to release of highly decomposed and modified organic matter. In contrast to the

bog, the peaty riparian zone is subjected to great water level changes and accelerated dry-wet cycles, which result in repeated aeration and enhanced decomposition (Singh et al., 2014).





### 4.1 Seasonal trends in DOM concentrations and quality

Although major changes in DOC quality occurred during individual discharge events, also seasonal patterns occurred at both sites, forest and bog. Main overall controls were temperature or vegetation period, respectively and differences in hydrology. DOC concentrations generally followed the vegetation period with low concentrations during snowmelt and spring and
highest DOC concentrations in early fall. This strong seasonal effect has been observed frequently and was ascribed to DOC production and solubility, but also enhanced litter decay by leaf fall in early fall (Christ and David, 1996; Singh et al., 2014; Wallin et al., 2015,). Nonetheless, especially at the forested site this seasonal trend was overprinted by hydrological events, which generated high DOC concentration peaks up to 45 mg L$^{-1}$ in fall due to rapid mobilization from hydrologically connected source areas.

All spectrofluorometric indices and PARAFAC components exhibited major changes during high discharge events and smaller changes by season. Thus they were mainly controlled by hydrologic (pre-)conditions, coinciding with little changes in DOM quality over the year as observed by (Perdrial et al., 2014). Due to predominance of Norway spruce, pulses of protein-like DOM from autumn leaf fall (Singh et al., 2014) would not be expected in our catchment.

### 4.2 Event DOM characteristics and DOM sources

#### 4.2.1 DOC concentrations

Organic-rich riparian zones are known to generate high DOC concentrations (Grabs et al., 2012). As a large part of the organic-rich upper soil layer is hydrologically connected and contributes to discharge only during events, high DOC concentrations at the forested site are due to the repeated flushing of peaty soils in the riparian zone. Additionally, upper organic layers of shallow hillslope soils may get connected via surface or surface near flow networks during such events.
Therefore, dry preconditions within the catchment facilitated high DOC concentrations during events of a certain magnitude when the upper soil layer gets hydrological connected. The DOC concentrations at the bog site were less sensitive to rain events and not as elevated during those events than at the forested site. Here, partly even decreasing concentrations were observed. Due to the usually high water level, rain events here do not connect additional DOM pools, but lead to dilution by surface flow or an exhaustion effect (Broder and Biester, 2015). We conclude that DOC concentration peaks during rain
events were mainly induced by peaty forests soils and not by bogs. Although the latter are strong C sources to the aquatic, they are less susceptible to rain events. DOC concentration trends of the bog site were further disentangled in (Broder and Biester, 2015).

#### 4.2.2 Snowmelt

During snowmelt DOC concentration at both sites were lowest. This has been reported elsewhere (e.g. Laudon et al., 2004;
Clark et al., 2008) and can be attributed not only to dilution by snow packs, but also to low microbial and plant activity. The absolute values of chromophore and fluorophore DOM were also lowest. Overall, spectrofluorometric indices point to rather





small molecular size, which were less aromatic. Additionally, PARAFAC components indicate a strong flush of labile, protein-like DOM. This characteristic was more pronounced at the forested site than at the bog site. The elevated export of a protein-like fraction, i.e. easily biodegradable DOM (Fellman et al., 2009), can be explained by less biotic demand and a domination of shallow flow paths during snowmelt bypassing large, strongly modified and aromatic DOM pools in the

subsurface (Fellman et al., 2009). Also, surface near freeze-thaw cycles during winter provide fresh DOM from microbial cell lysis and root mortality, which is not utilized due to the low productivity (Haei et al., 2012, Fellman et al., 2009). Differences between the two sites were more evident in other PARAFAC components. At the forested site a large fraction of humic-like fluorescence was attributed to C3%, while C4% was absent. C3% only increased strongly at the forested site during rain events, while C3% remained constant at the bog site. This suggests that component C3% is predominantly

sourced in the upper, organic layer of the forest soil and is only mobilized during rain events, when these surface near layers get hydrologically connected, even during snowmelt. As this component increases strongest of all identified components, higher DOC concentrations during rain events should be mainly caused by a connection of surface-near organic-rich layers to the streams, irrespective of the season. In our study an increase of C3% coincide with lower $SUVA_{254}$ and higher $S_R$ and FI values. This points to a microbially modified, recalcitrant, but less aromatic DOM fraction of smaller apparent molecular

size, confirming previous descriptions of a largely similar fluorophore (e.g Fellman et al., 2008; Singh et al., 2014). Overall, the snowmelt DOM can be described as smaller, less chromophoric, less aromatic and more biodegradable due to a higher protein-like fraction, which is more pronounced at the forested site and has its source in the upper soil layer.

### 4.2.3 Drought

A drought period during summer 2013 caused a strong decrease in the bog water level down to 35 cm depth and

exceptionally low discharge representing pronounced baseflow conditions. This dry period induced concomitant changes in DOM quality in stream discharge. While DOC concentrations at both sites continuously increased, aromaticity and apparent molecular size decreased. While FI at the bog site remained constant (around 1.55), FI at the forested site increased during drought up to 1.75 indicating a greater fraction of microbial derived, strongly modified DOM. Under these drought conditions, the indices congruently picture a less aromatic, smaller, and more microbial DOM at the peaty forest soils

compared to the bog site and to the rest of the record. This can be explained by a higher contribution of shallow groundwater and decreasing discharge through and DOM contributions from the peaty surface layers of the riparian zone. Due to the adsorption of larger, more aromatic compounds to mineral phases, groundwater DOM is typically of smaller molecular size and less aromatic (Meier et al., 2004; Inamdar et al., 2012). Also for the bog site increasing C1% and bog-derived humic-like C4%, as well as decreasing molecular size hint to a DOM source change toward deeper peat layers. Summarizing the

summer drought fingerprint, DOM during this period approaches characteristics observed for shallow groundwater at the forested site and resembles DOM from deeper peat layers at the bog site. This caused a change in DOM quality to smaller, more microbial and less aromatic components.




### 4.2.4 Rain events

The three sampled rain events distributed over spring and fall clearly differed with respect to observed changes in DOM quality. The spring events can be differentiated between dry (first event) and wet (second event) preconditions, while the fall event occurred again under dry preconditions. The main difference between these two preconditions was a higher DOC

concentration under dry preconditions at both sites. While this difference in DOM concentration could be attributed to a dilution effect under wet preconditions at the bog site, also notable differences in DOM quality occurred: The non-fluorescent DOM fraction was clearly elevated following dry preconditions. This indicates that under dry preconditions a specific DOM fraction, which cannot be separated by fluorometric indices, is exported compared to wet preconditions. As non-fluorescent DOM is probably more easily degradable DOM like organic acids or products of biotic activity, this might

be a flushing effect, when upper soil layer gets hydrological connected after prolonged aeration, decomposition and concomitant enrichment of potentially mobile DOM, while under wet preconditions this DOM fraction gets exhausted. As this flushing effect also occurred at the dry fall event, the export of this DOM fraction may be mainly attributed to the hydrologic preconditions and not to the low demand of labile DOM in early spring, as has been suggested elsewhere (Fellman et al., 2009). This would further imply that focusing on the protein-like C5% as proxy for BDOM (Fellman et al.,

2008) neglects a further labile DOM fraction, which might serve as nutrient source downstream and does not correlate with protein-like fluorescence.

A major seasonal difference between spring and fall event is, that there were no elevated protein-like DOM exports in fall. However, elevated contributions of protein-like fluorescence as described before (Singh et al., 2014) were ascribed to leaf fall at that time of the year. As in our catchment only coniferous trees occurred, this may not be observed in this study.

However, the contribution of the non-fluorescent fraction in fall was even higher than in spring. This might be indicative of higher biotic activity, generating small, non-fluorescent molecules, and demonstrates the limitations of the spectrofluorometric approach of DOM characterization.

At the forested site SUVA$_{254}$, HIX and S$_R$ indicated a lower aromaticity and smaller molecules during all rain events. This is confirmed by an increasing FI up to 1.74, meaning a shift to more microbial DOM. The DOM was even of smaller apparent

molecular size and less aromatic even though trends of the indices were partly reversed in fall. This reversal was due to the extreme dry preconditions and prevailing baseflow with a high microbial derived DOM fraction, primarily originating from groundwater and deep peat. With the onset of a rain event the organic soil layers get hydrologically connected, leading to a decrease of microbially, shallow groundwater derived compounds and an increase in more aromatic DOM originating from the strongly humified organic matter of the peaty layers. Taking together the trend of the humic-like C1% at the forested site

during events and drought period reveals that this component probably represent a rather microbial processed DOM fraction from groundwater, while humic-like C2% and C3% largely contribute to DOM under high discharge conditions and therefore represent a DOM sourced in the upper soil layers. In contrast to other studies (Hood et al., 2006, Inamdar et al., 2011), a general increase of aromatic DOM during rain events at the forested site was not recorded, but an increase in humic





and microbial DOM indicating that the humic-like C3% is also microbial derived DOM, but from another source. However, under wet preconditions the decrease in $SUVA_{254}$ was less distinct and C2%, indicative of a shift to rather aromatic structures, contributed stronger to fDOM than under dry preconditions. This indicates a stronger aromatic contribution to DOM export from upper soil layer under wet preconditions at the forested site, while under dry preconditions longer aeration may yield more modified, less aromatic DOM that has pooled up during drought.

At the bog site under wet preconditions much less DOC was exported than at the forested site indicating an exhaustion, as well as a dilution effect of the surficial DOM pool here (Broder and Biester, 2015). Hydraulic conductivities at greater depths, where high DOM concentrations prevail, are presumably too low for rapid mobilization. Compared to changes at the forested site, aromaticity only moderately shifted during rain events and over seasons. However, even though aromaticity was not elevated at the bog site, $S_R$ indicated DOM of a rather large molecular size being mobilized under wet preconditions. Also the protein-like component was highest at the wet spring event, which is explained by flushing of fresh biotic material in the surface layer. Moreover, during this event overland flow is very likely (Broder and Biester, 2015), which might further leach larger polymers of proteins, cellulose or polysaccharides from the living biomass. Also, (Fellman et al., 2009) reported an increase in protein-like DOM export and related this to lower residence times and low biotic demand in a wetland catchment. Unfortunately, $S_R$ was not monitored in that study. The bog specific humic-like C4% component only moderately decreased during rain events and increased over summer drought at the bog site. This dynamic in C4% presumably describes a component from deeper peat layer, which is constantly exported over the year and gets diluted by upper surface or surface-near export during high discharge events. Therefore, this component may be used as tracer for deep bog porewater and the observed dilution effect clearly points out that bogs do not primarily drive variations in DOM loads of streams.

Summarizing dynamics during events, DOM quality changes reflect different contributions of DOM pools depending on hydrologic preconditions and season. Fresh and labile DOM was exported during spring events at both sites, especially under wet preconditions at the bog site. Even though aromaticity of DOM in the studied catchment was high, events showed an increase in microbial or strongly microbially altered DOM. However, PARAFAC components show that this assumed microbial component is not only sourced in shallow groundwater, but that there is an additional microbial DOM pool in the upper soil layer that is especially mobilized under dry preconditions. Comparing the two sites our results demonstrate that not only major dynamics in DOM quantities, but also variability in DOM quality was mainly driven by the forested site, i.e. by shallow peaty soils with stronger variations in water tables and thus hydraulic connectivity of the different layers.

## 5 Conclusion

Overall, changes in DOM quality were mainly induced by hydrologic conditions, which points out the importance of high resolution studies and consideration of high discharge events, which not only generate highest DOC concentrations, but export different DOM pools with different chemical properties and fate in aquatic systems. Main variability in stream DOM concentration and characteristic was thereby generated at the forested site with peaty riparian zones and not at the bog site.



The export of labile, protein-like DOM was specific for snowmelt and for spring events after wet preconditions at this study site. Those DOM compounds might serve as important nutrient source in the aquatic system. Nevertheless, not only during spring events, but also in fall a non-fluorescent DOM fraction of small apparent molecular size was exported that may be of similarly high bioavailability. During drought periods DOM export was limited to deeper peat layer at the bog site and shallow groundwater at the forested site as could be tracked by a specific strong microbial DOM signature originating from long DOM residence times in the soil and peat. At events with wet preconditions additional surface-near DOM pools were connected due to increasing water levels in the catchment. Next to aromatic DOM compounds, a strong surface-near microbial DOM fraction was exported during those events, as could be tracked by specific PARAFAC components. While at the bog site a dilution effect of DOM concentration sets in under wet preconditions the forested site generated highest DOM concentration peaks under wet preconditions.

The PARAFAC modeling of different DOM components proofed a useful tool to track export dynamics of different DOM pools under different seasonal and hydrologic conditions, which could have not been resolved by the spectrofluorometric indices alone. Moreover, our study demonstrates the need for approaches tracking DOM sources to understand DOM export dynamics, while approaches based solely on hypothetic hydrological compartments, such as surface flow, soil water and groundwater, may be too simplistic. This understanding of how different DOM pools gets exported might become even more important in view of future changes in the hydrologic regime due to climate change.

**Competing interests**

The authors declare that they have no conflict of interest.

**Acknowledgments**

This work was funded by the NTH graduate school "GeoFluxes" of the Federal State of Lower Saxony, Germany. We are grateful to the the Nationalpark Harz for giving access to the site. The authors thank Adelina Calean, Petra Schmidt, and Julian Fricke for help with lab and field work. UV-VIS and fluorescence spectroscopy were carried out in the laboratory of the Institute of Landscape Ecology at the University of Münster. We thank Johan Rydberg for helpful contributions and Christian Blodau for scientific and financial support.

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



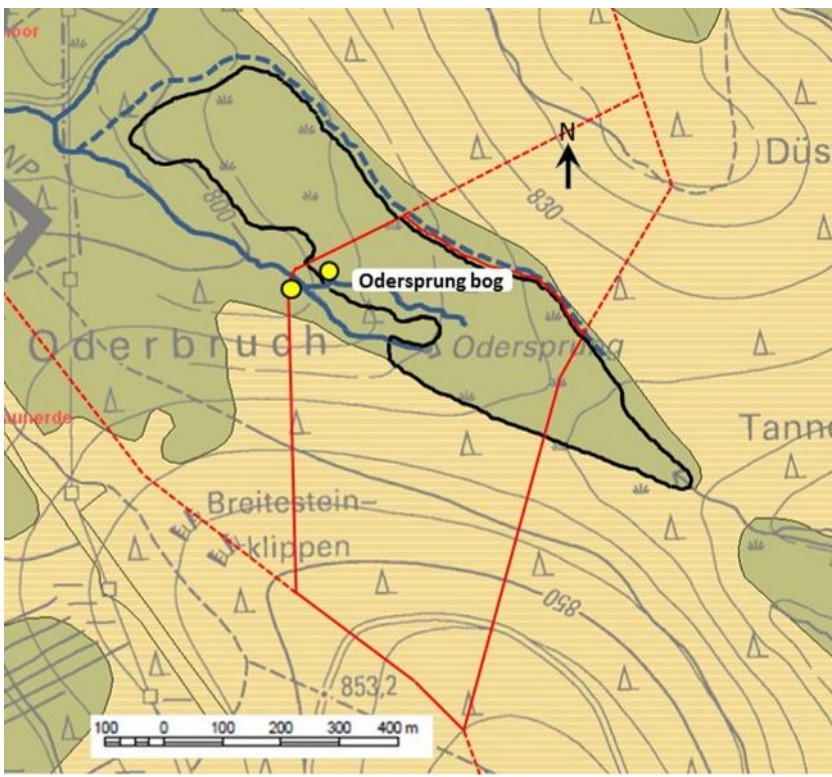

**Figure 1: Location of the study area in the Harz Mountains, Germany. Red lines indicate the catchment boundaries, yellow circles represent the discharge monitoring spots. Green areas indicate peatland or peaty soils, beige-colored areas outline mineral cambic podzol soils.  Map Source: NIBIS mapserver, Lower Saxony authority for mining, energy and geology (LBEG).**





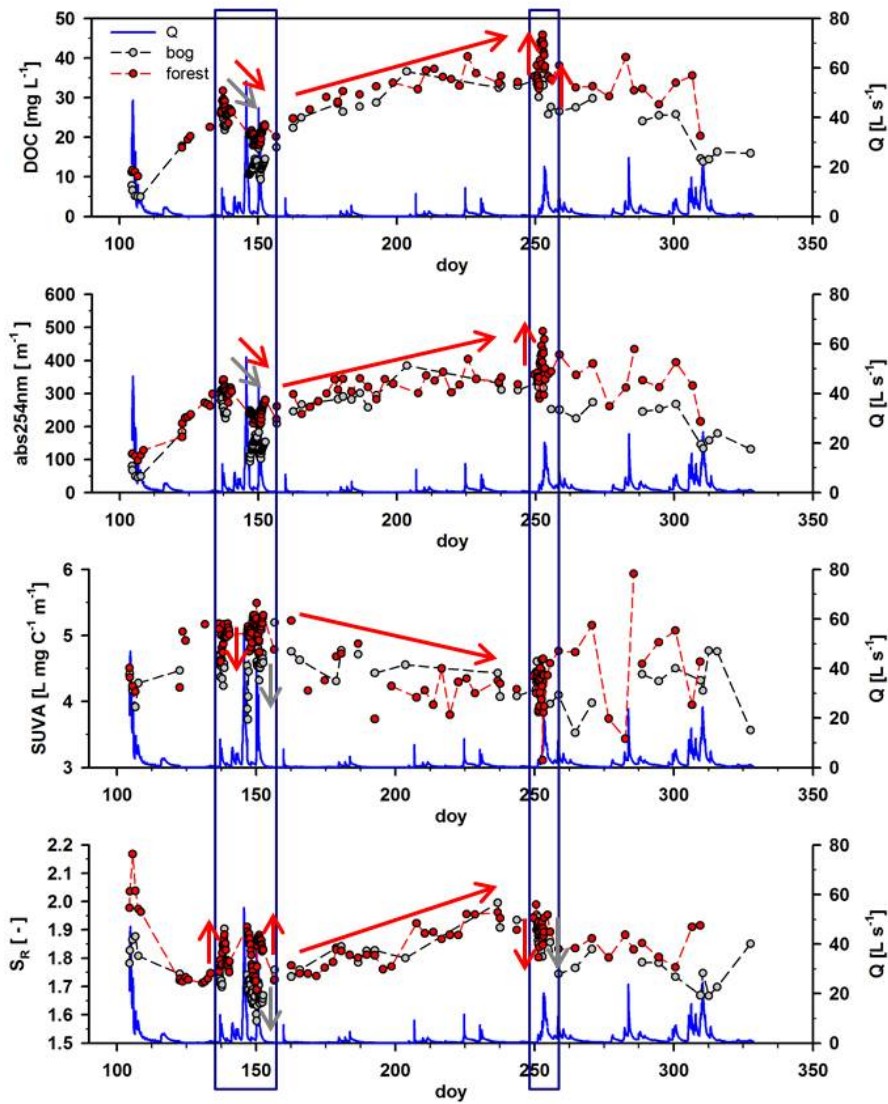

**Figure 2: Annual records of DOC concentrations, abs$_{254nm}$, SUVA$_{254}$, and S$_R$ from top to bottom (DOY - day of the year) in 2013. The blue line represents the bog discharge (Q). Grey circles represent the bog site, red circles the forested site, while arrows indicate trends during rain events and summer drought at the different sites. Sampled rain events in spring and fall are highlighted by blue boxes.**



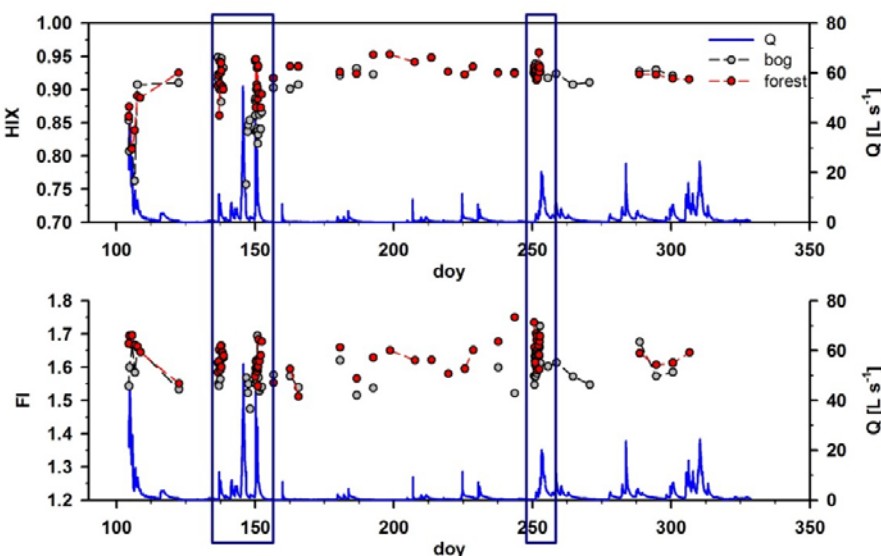

**Figure 3: Annual record of the humification index (HIX) and fluorescence index (FL) (DOY - day of the year) in 2013. The blue line represents the bog discharge (Q). Grey circles represent the bog site, red circles the forested site. Sampled rain events in spring and fall are highlighted.**

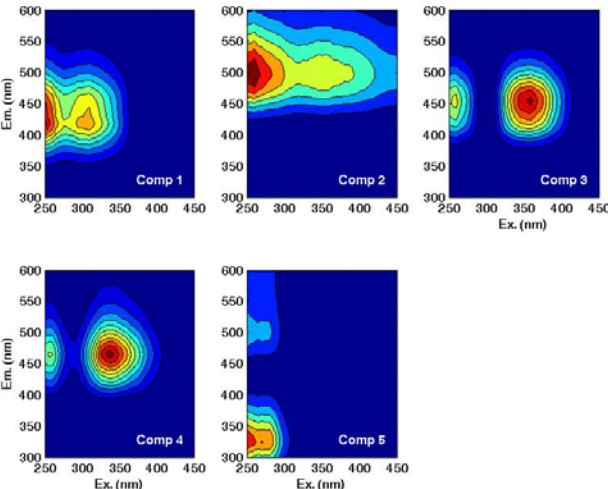

**Figure 4: Characteristic EEMs of all five modeled PARAFAC components. For further component description see section 3.1.3.**





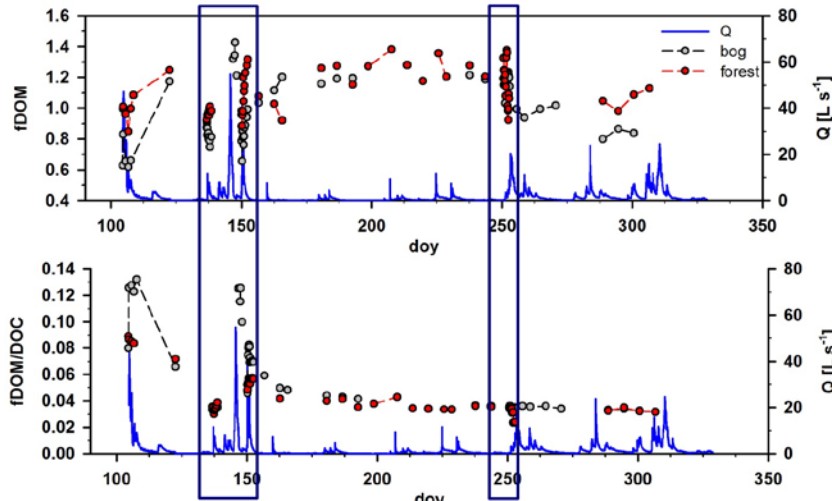

**Figure 5: Sum of all Fmax PARAFAC components values (fDOM) and fDOM to DOC concentration (in mg L$^{-1}$) ratio against day of the year (DOY) in 2013. Blue line represents the bog discharge (Q). Grey circles represent the bog site, red circles the forested site. Sampled rain events in spring and fall are highlighted.**





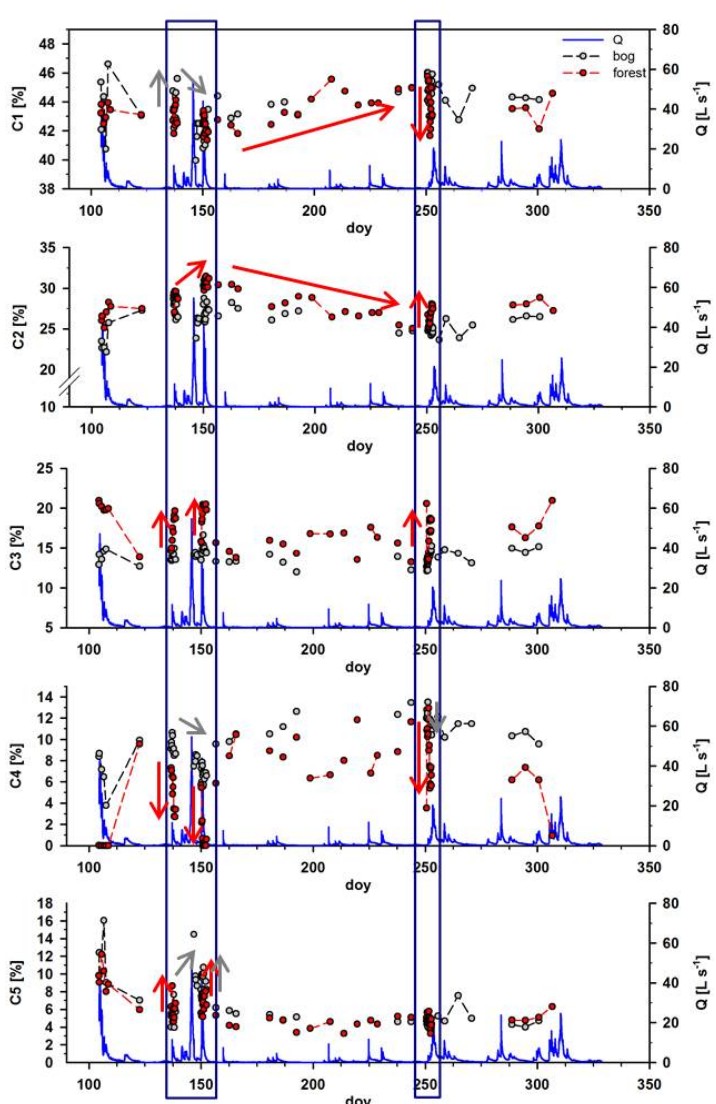

**Figure 6: Percentage of PARAFAC components C1-C5. Blue line represents the bog discharge (Q). Grey dots indicate bog site values, red dots forest site values. Sampled rain events in spring and fall are highlighted.**




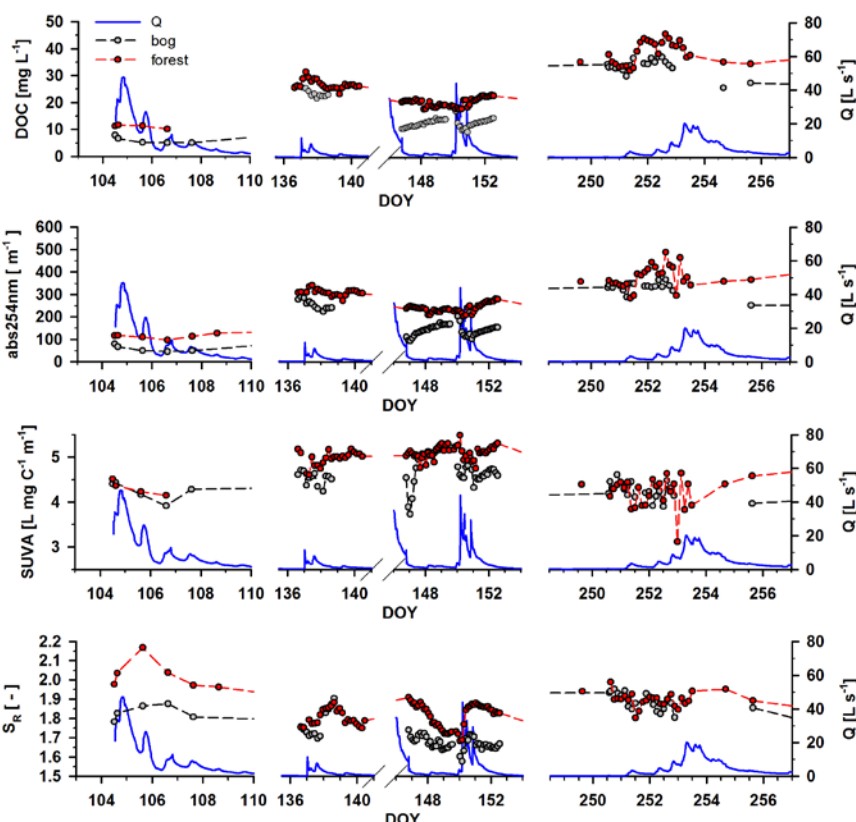

**Figure 7: Event DOC concentrations, abs$_{254nm}$, SUVA$_{254}$, and S$_R$ from top to bottom during snowmelt (DOY 104-110), spring events (DOY 136-155) and fall event (DOY 247-258). Blue line represents the bog discharge (Q). Grey dots indicate bog site values, red dots forest site values.**





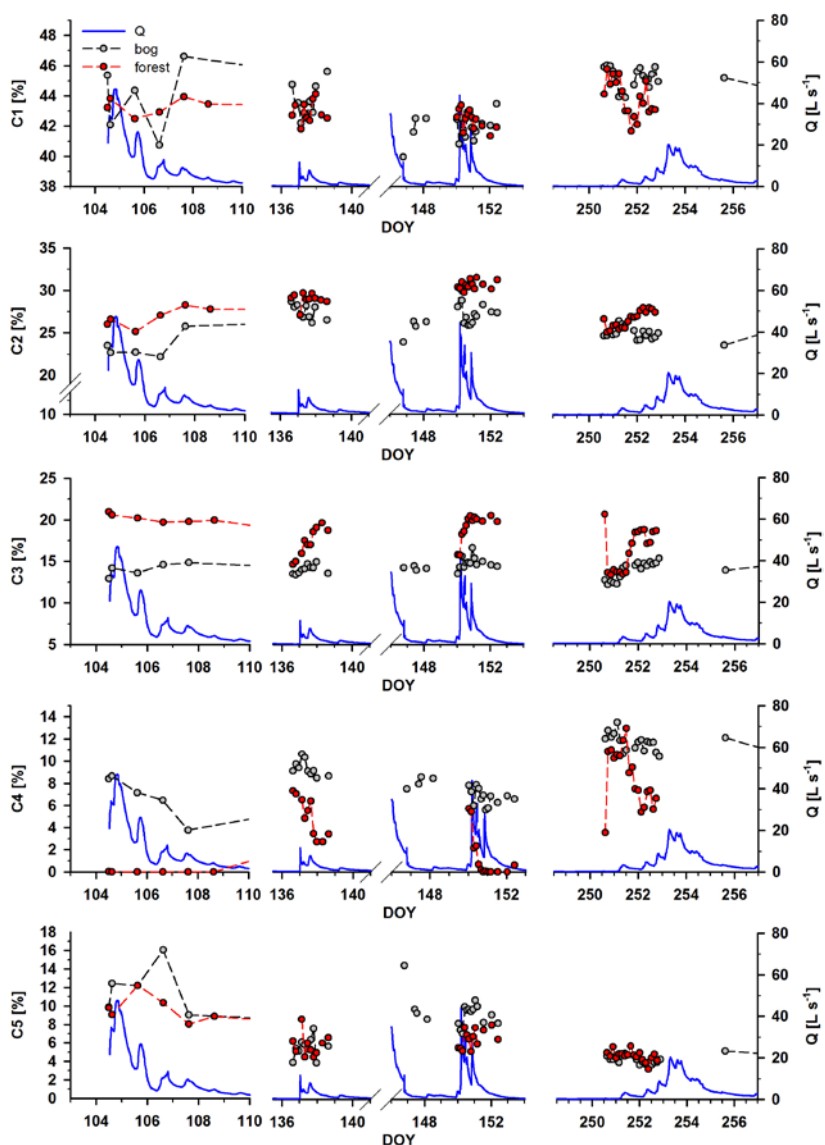

**Figure 8: Percentage of PARAFAC components during snowmelt (DOY 104-110), spring events (DOY 136-155) and fall event (DOY 247-258). Blue line represents the bog discharge (Q). Grey dots indicate bog site values, red dots forest site values.**