# Peer review of "Changes in dissolved organic matter quality in a peatland and forest headwater stream as a function of seasonality and hydrologic conditions"

_Hydrology and Earth System Sciences, 2016_

## Referee Comment (RC1) · Anonymous Referee #1 · 7 Oct 2016

General comments

This manuscript aims at disentangling stream DOM sources and processes over the course of a year. It utilyzes spectrofluorometric methods that allow for a high throughout of samples. I very much like the approach of the manuscript and the dataset presented is worth a paper in HESS. So I can confirm a excellent scientific significants. Alternatively a publication in Biogeosciences would fit as well as the manuscript elucidate the interactions between hydrology and biogeochemistry.

In general the manuscript is well written, references are appropriate and figures are sufficient (see details below). The objectives could be improved by a clearer focus and formulation of hypotheses or research questions. The major drawback is the only fair scientific and presentation quality that needs to be improved. My major criticism

is the simpified verbal description of the results that lacks quantitative assessments of concentrations/ component levels, seasonality and event behavior. Also, a quantitative statistical analysis on major governing factors such as temperature is needed. Here I see much room of improvement that would make the well written conclusions much more stronger and compelling.

Specific comments and technical corrections

Abstract:

For me, it reads a bit to technical and focussed on the results. I would like to see terms and variables like "C2%" to be reduced in favor of a stronger focus on discussion and implications (expanding the messages now in the last sentence).

Introduction

P2Line5: The first sentence need a reference.

P2Line 7f: This also needs a reference.

Check also for this paper: Ledesma JLJ, Grabs T, Bishop KH, Schiff SL, Kohler SJ (2015) Potential for long-term transfer of dissolved organic carbon from riparian zones to streams in boreal catchments. Global Change Biology, 21, 2963-2979

P2L34: Change citation parenthesis.

P3Line19: Better define what you mean with changes by adding space and time! Do you mean surface water DOM at the catchment outlet or variability within the study site – do you mean seasonal changes, daily fluctuations...: Something like "spatiotemporal dynamic of DOM quality over the course of one year" ...

P3Line23: Two times starting a sentence with "on the other hand"

P3Line31: This last section is not well integrated with the objectives in the section before: I suggest restructuring the objectives a bit starting from P3Line23; there is more

or less everything there: What is the general aim? What are specific questions you want to answer/ Hypotheses to follow? Where do you perform this research (references on previous studies) and why there? What are the methods you want to apply? What are discussion/ conclusion and implications aiming at?

Materials and methods

P4Line1-14: I am not convinced of the study site description in combination with Fig. 1: I suggest to show a soil map or something comparable to better see the position of the bog and the differences between the two sampling locations.

P4Line15: "Discharge sampling" sounds strange – just water sampling?

P4L15ff: Please state the number of different types of samples and the temporal resolution of the discharge measurements.

P4Line27: Unit of absorption?

P4Line29f: Superscript in units!

Results

Fig. 2: What do you mean by "trends during rain events"? I suppose the concentration trends?

P5Line29f: Please put that statement in numbers (e.g. CV or standard deviation of seasonal concentrations and storm event concentrations).

P5Line30f: You mean the highest recorded concentrations during the entire study period? This is not clear from this sentence.

P6Line1: Again – put that into numbers (e.g. mean or median concentrations)

P6Line23ff: I struggle with the interpretation of the components as a "too early" discussion. Maybe a introduction sentence on that?

P7Line19ff: Again, I have problems with this type of interpretation in the result section.

[Figure]

This more belongs to the discussion chapter.

In general I am not convinced of the discussion of seasonal trends: For a seasonality assessment I would expect a more in depth evaluation of seasonal min and max and potential controls: e.g. Is the seasonality in line with water temperature (which is likely close to soil water temperature), air temperature, light intensity... So, better quantify and describe the seasonality!

Discussion

P9Line 6f: This type of statement belongs to the end of discussion/ conclusions!

---

## Referee Comment (RC2) · Anonymous Referee #2 · 11 Oct 2016

The study of Tanja Broder, Klaus-Holger Knorr and Harald Biester used spectrofluorometric indices to relate the quality of DOM to potential processes and hydrological sources of DOM generation. I recommend addressing the following issues in a revised manuscript:

The study is intended to elucidate the DOM quality and dynamics in two contrasting catchments. The authors expect "water level, temperature and precipitation to be the main controls on stream DOM . . .". I would acknowledge if the authors provide a more specifically defined working question/hypothesis along which the results and discussion can be organized. How can the DOM of both catchments differ?

Samples were taken by an automated sampler in a six day interval and the filtered samples were stored at 4°C in the dark. How long were the samples stored until analysis?

Although filtered, the samples were probably not sterile. Bacteria can degrade organic carbon even at low temperatures. Have the authors tested if the DOC composition remained unchanged?

UV absorption characteristics were used as indicators of DOM composition (absorbance at 254 nm, spectral slope ratio). Besides DOM, however, dissolved iron exhibits significant UV absorption. The dynamics of iron is also related to hydrology, i.e. flood events can be associated with high iron concentrations. Changes in UV absorption may reveal changes in DOM quality but at the same time they can reflect different contributions of dissolved iron. How the authors think about that?

The fluorescence index (FI) differentiates between vascular plant derived (FI 1.3 – 1.4) and microbially derived (1.7-2.0) DOM. FI values >1.7 were interpreted as microbial DOM (p. 5 lines 15-20). Can the authors exclude that non-vascular plants (mosses) contributed to DOM generation? Later in the manuscript a contribution of Sphagnum is specifically discussed with respect to SUVA but not to FI (p. 9 lines 15-31).

In the results and also in the discussion sections, fluorescence indices and absorption values were often reported to be higher or lower when sites or situations were compared. However, I missed statistics providing a little more confidence if these differences are significant. I recommend including a table/figure, summarizing the main results as well as the levels of significance. This table/figure along with a hypothesis can be used to guide the reader through the results and discussion. I found it less convenient to work through the detailed description of the results. The figures 2, 3, 5, 6, 7 and 8 look more or less similar, which makes it not easy to keep the most important results in mind.

Further comments The title: Please refer more directly to the outcome of the study p. 1 lines 22-24: The concluding sentence of the abstract "Our study demonstrated that DOM export dynamics are not only a passive mixing of different hydrological sources, but. . ." is unclear. Now the reader expects a statement why it is not a passive mixing or how the process can be characterized instead. However, it is only concluded that "...assessing DOM quality can greatly improve our understanding...". Please include here the most relevant result that improved our understanding. The sources and the quality of DOM appeared to be highly variable within events depending on runoff generation. What are the consequences for sampling/monitoring programs?

p. 2 line 28: "aromatic or humic", is there a difference? p. 2 line 33: groundwater DOM is of smaller size and mostly of microbial origin, please include a reference p. 4 line 29: The near UV includes light from 300 – 400 nm, 254 nm is in the middle UV range p. 7 line 9: Figure 5 instead of Figure 6? p. 8 lines 10 and 16-17: I had problems to relate these statements to the figures p. 9 lines 6-9: A confirmation of the suitability of fluorometric indices appears difficult without independent methods (e.g. isotopes, mass spectrometry). It is problematic to conclude that an increase in aromaticity is caused by an increase in apparent molecular size if the latter is not measured. p. 14 line 1: "The export of labile, protein-like DOM was specific..." I suggest being more cautious with characteristics of DOC that have not measured directly (e.g. "labile"). See also P. 9 line 5 "...specific strong microbial DOM signature...".

In Figures 1, 3, 5 and 6, the individual symbols in highlighted boxes (events) were difficult to distinguish.

---

## Author Comment (AC1) · 25 Nov 2016

**Author's Response to Referee #1**

We would like to thank the Referee for the helpful and constructive comments, which will greatly improve the manuscript. Our detailed responses to the comments are listed below.

*# The objectives could be improved by a clearer focus and formulation of hypotheses or research questions. The major drawback is the only fair scientific and presentation quality that needs to be improved. My major criticism is the simpified verbal description of the results that lacks quantitative assessments of concentrations/ component levels, seasonality and event behavior. Also, a quantitative statistical analysis on major governing factors such as temperature is needed.*

Response: In our revised manuscript we will have reworked the last section of our introduction to clarify our objectives and research question. Additionally, we add statistical analysis (descriptive, as well as correlations and tests of significant differences) to our data to strengthen our statements. These issues will be addressed in further detail in the specific comments of this response (see below).

*# Specific comments and technical corrections*

*# Abstract: For me, it reads a bit to technical and focussed on the results. I would like to see terms and variables like "C2%" to be reduced in favor of a stronger focus on discussion and implications (expanding the messages now in the last sentence).*

Response: As suggested by the referee we will adjust the focus of the abstract rather to discussion and conclusions and will shorten the result section. We will revise the abstract also in view of the comments of referee #2. We will include major outcomes of the study like the high variability of DOM quality due to hydrologic conditions, which makes it necessary to cover discharge events during monitoring studies. We will outline the differences between both studied catchments (bog/forested riparian zone) and shortly discuss the suitability of spectrofluorometric indices to track DOM origin and dynamics in this headwater stream.

*# Introduction: P2Line5: The first sentence need a reference.*

Response: We will include a reference in the revised version of the manuscript (Mulholland 2003).

*# P2Line 7f: This also needs a reference. Check also for this paper: Ledesma JLJ, Grabs T, Bishop KH, Schiff SL, Kohler SJ (2015) Potential for long-term transfer of dissolved organic*

*carbon from riparian zones to streams in boreal catchments. Global Change Biology, 21, 2963-2979*

Response: We will include a reference in the revised version of the manuscript as well and thank the referee for his/her literature suggestion, which slipped our thorough literature search. We intend to use that study as reference at another point. We will refer to e.g. Fellman et al. 2009, Perdrial et al. 2014 and Wallin et al. 2015 at this point instead.

*# P2L34: Change citation parenthesis.*

Response: We will change that in the revised manuscript.

*# P3Line19: Better define what you mean with changes by adding space and time! Do you mean surface water DOM at the catchment outlet or variability within the study site – do you mean seasonal changes, daily fluctuations...: Something like "spatiotemporal dynamic of DOM quality over the course of one year" ...*

Response: We will rephrase that sentence to avoid misunderstanding. The study aim described here is to elucidate different spatiotemporal dynamics in DOM quality over a year in a headwater stream, as the reviewer suggested. Those dynamics are supposed to be controlled by the bog and forested peaty riparian zone catchment.

*P3Line23: Two times starting a sentence with "on the other hand"*

Response: We used the English phrase "on the one hand… On the other hand…" here, which we think is used correctly in our manuscript.

*# P3Line31: This last section is not well integrated with the objectives in the section before: I suggest restructuring the objectives a bit starting from P3Line23; there is more or less everything there: What is the general aim? What are specific questions you want to answer/ Hypotheses to follow? Where do you perform this research (references on previous studies) and why there? What are the methods you want to apply? What are discussion/ conclusion and implications aiming at?*

Response: We agree with the referee and will revise this section of the manuscript in view of the mentioned deficiencies and comments of referee #2. We will clarify our objectives/hypothesis in view of these points: I) To test if spectrofluorometric indices can be used to track origin and dynamics of DOM. II) That DOM quality is highly variable in a headwater stream depending on hydrologic conditions and season. We expect short-term

DOM quality changes due to high discharge events, which cause changes in hydrologic connectivity of different DOM pools and possible leachate effects of labile DOM during strong rain events. This short-term pattern is expected to be overlain by seasonal DOM changes due to changes of DOM production and consumption over the year. III) We expect general differences in DOM quality between the bog and forested riparian zone catchment. The riparian zone is characterized by accelerated water level fluctuations and a nutrient-richer vegetation than at the bog site, which leads to the hypothesis that DOM quality is strongly affected by changes in hydrologic conditions and is more labile than at the bog site. We will point out the suitability of this particular catchment as the stream origins within the bog and enables us to retrieve a exclusively bog derived DOM signal within this headwater stream. We will also refer to our previous study at this catchment and the applied spectrofluorometric methods, which are $SUVA_{254}$, $S_R$, FI, HIX and PARAFAC modeling of excitation emission matrices.

*# Materials and methods: P4Line1-14: I am not convinced of the study site description in combination with Fig. 1: I suggest to show a soil map or something comparable to better see the position of the bog and the differences between the two sampling locations.*

Response: The provided Figure 1 is actually a soil map. We admit that the figure might not sufficiently highlight the information, which is important to this study. To make the figure more comprehensible we will extend the figure caption and modify the map, clarifying the declaration of soil type distributions and sampling spots.

*# P4Line15: "Discharge sampling" sounds strange – just water sampling?*

Response: We agree with the referee and will rephrase that expression.

*# P4L15ff: Please state the number of different types of samples and the temporal resolution of the discharge measurements..*

Response: We will include this information in the revised manuscript. In short, we took 30 grab samples, 44 water samples by the automated water sampler and 191 samples during high frequency storm event sampling. Discharge was recorded in a 10 minute time interval.

*# P4Line27: Unit of absorption?*

Response: The missing unit of absorption is $m^{-1}$. We will correct that in the revision.

*# P4Line29f: Superscript in units!*

Response: We will edit this mistake.

*# Results: Fig. 2: What do you mean by "trends during rain events"? I suppose the concentration trends?*

Response: We agree that this is a poor expression. We will rephrase it to "trend of concentration" or "trend of quality indices", respectively.

*# P5Line29f: Please put that statement in numbers (e.g. CV or standard deviation of seasonal concentrations and storm event concentrations).*

Response: We pick up the referees comment and will strengthen our statements with a table (Table 1 in a revised manuscript) of descriptive statistics including mean, median, standard deviation and minimum and maximum values. In order to present the data more comprehensively we also consider to present boxplots for parts of the data.

*# P5Line30f: You mean the highest recorded concentrations during the entire study period? This is not clear from this sentence.*

Response: The referee is right, the statement refers to the entire study period. We will rephrase that sentence for clarification.

*# P6Line1: Again – put that into numbers (e.g. mean or median concentrations)*

Response: As described to a previous comment we will include a new table, which will provide the requested numbers.

*# P6Line23ff: I struggle with the interpretation of the components as a "too early" discussion. Maybe a introduction sentence on that?*

Response: We included the interpretation of the components in the results after careful consideration and with regard to existing literature to facilitate reading and help the reader understand the common use of these indices. Therefore, we did not consider this as a real "discussion" here. If we include this information only in the discussion section, the result section describing characteristics of different rain events would more difficult to follow, e.g. as the description of components would remain very technical and abstract. Furthermore the focus of the discussion would be less clear, if we introduce the common interpretation of

components only there. From our point of view this interpretation is thus rather a description of the modelled components, in terms of 'results', backed up by literature data (which is admittedly unusual for a normal result section). Such description is found also in the results or method sections in other studies. Nevertheless, to address this unease of an early discussion we will include an introducing sentence.

*# P7Line19ff: Again, I have problems with this type of interpretation in the result section. This more belongs to the discussion chapter.*

Response: We will revise that mentioned section (see above) and put the interpreting statements of the last two sentences (P7Line19ff) to the discussion section. We will limit the description solely to the components and their contributions (referring to studies identifying similar components) without assigning a source of these components here, but putting this source assignment into the discussion.

*# In general I am not convinced of the discussion of seasonal trends: For a seasonality assessment I would expect a more in depth evaluation of seasonal min and max and potential controls: e.g. Is the seasonality in line with water temperature (which is likely close to soil water temperature), air temperature, light intensity... So, better quantify and describe the seasonality!*

Response: We thank the referee for this constructive comment. We agree that statistical indications are helpful to underline our statements in regard to seasonality. We will include a description of minimum and maximum values in the revised version. Unfortunately, we did not measure soil temperature and only occasionally water temperature, which limits options to further evaluate seasonality effects as suggested. We chose to use air temperature as a proxy for seasonality, however further parameters from a meteorological weather station (such as light intensity) would be available in case the reviewer suggests to include more data here.

*# Discussion: P9Line 6f: This type of statement belongs to the end of discussion/ conclusions!*

Response: We agree with the referee and move this point in the revised manuscript.

References:

Fellman, J. B., Hood, E., D'Amore, D. V., Edwards, R. T., and White, D.: Seasonal changes in the chemical quality and biodegradability of dissolved organic matter exported from soils to streams in coastal temperate rainforest watersheds, Biogeochemistry, 95, 277–293, 2009.

Mulholland, P. J.: Large-Scale Patterns in Dissolved Organic Carbon Concentration, Flux and Sources, in: Aquatic Ecosystems: Interactivity of Dissolved Organic Matter, Findlay, S. E. G., and Sinsabaugh, R. L. (Eds.), Academic Press/Elsevier, San Diego, London, 139–160, 2003.

Perdrial, J.N., McIntosh, J., Harpold, A., Brooks, P. D., Zapata-Rios, X., Ray, J., Meixner, T., Kanduc, T., Litvak, M., Troch, P. A. and Chorover, J.: Stream water carbon controls in seasonally snow-covered mountain catchments: impact of inter-annual variability of water fluxes, catchment aspect and seasonal processes, Biogeochemistry, 118, 273-290, doi:10.1007/s10533-013-9929-y, 2014

Wallin, M. B., Weyhenmeyer, G. A., Bastviken, D., Chmiel, H. E., Peter, S., Sobek, S. and Klemedtsson, L.: Temporal control on concentration, character, and export of dissolved organic carbon in two hemiboreal headwater streams draining contrasting catchments, Journal of Geophysical Research: Biogeosciences, 120, 832-846, doi: 10.1002/2014JG002814, 2015

---

## Author Comment (AC2) · 25 Nov 2016

**Author's Response to Referee #2**

We would like to thank the Referee for the helpful and constructive comments, which will greatly improve the manuscript. Our detailed responses to the comments are listed below.

*# Specific comments:*

*# The study is intended to elucidate the DOM quality and dynamics in two contrasting catchments. The authors expect "water level, temperature and precipitation to be the main controls on stream DOM…". I would acknowledge if the authors provide a more specifically defined working question/hypothesis along which the results and discussion can be organized. How can the DOM of both catchments differ?*

Response: Also in line with referee #1 we will rephrase our objectives in the introduction to clarify our hypothesis. Main objectives/hypotheses were: I) To test if spectrofluorometric indices can be used to track origin and dynamics of DOM. II) That DOM quality is highly variable in a headwater stream depending on hydrologic conditions and season. We expect short-term DOM quality changes due to high discharge events, which cause changes in hydrologic connectivity of different DOM pools and possible leachate effects of labile DOM during strong rain events. This short-term pattern is expected to be overlain by seasonal DOM changes due to changes of DOM production and consumption over the year. III) We expect general differences in DOM quality between the bog and forested riparian zone catchment. The riparian zone is characterized by accelerated water level fluctuations and a nutrient-richer vegetation than at the bog site, which leads to the hypothesis that DOM quality is strongly affected by changes in hydrologic conditions and is more labile than at the bog site. Additionally, we will review our result and discussion section in term of hypothesis consistency.

*# Samples were taken by an automated sampler in a six day interval and the filtered samples were stored at 4°C in the dark. How long were the samples stored until analysis? Although filtered, the samples were probably not sterile. Bacteria can degrade organic carbon even at low temperatures. Have the authors tested if the DOC composition remained unchanged?*

Response: The storage time of our samples depended upon the type of measurement. DOC concentration and UV-Vis analytics were conducted at the TU Braunschweig. We could therefore measure samples more or less immediately (within a week). Immediate fluorescence measurements were impossible, as those were conducted at the University of Münster, which necessitated measurement campaigns in about three months intervals. To address the problem of conservation we tested UV-VIS changes, which were inconspicuous over that time-period. Furthermore, due to the longtime experience of the ecohydrology group in Münster in fluorescence measurements we came to the conclusion that we

produce less analytic artefacts by storing the samples at low temperatures over that time period than freeze them for conservation. We also refer to following literature: Santos et al. 2010, Hudson et al. 2009, Graeber et al. 2012. There it also is recommended that samples should be frozen, if storage time is longer than one year.

*# UV absorption characteristics were used as indicators of DOM composition (absorbance at 254 nm, spectral slope ratio). Besides DOM, however, dissolved iron exhibits significant UV absorption. The dynamics of iron is also related to hydrology, i.e. flood events can be associated with high iron concentrations. Changes in UV absorption may reveal changes in DOM quality but at the same time they can reflect different contributions of dissolved iron. How the authors think about that?*

Response: This is an important point. We are aware of biases of UV-VIS and fluorescence DOM characterization by Fe concentration or pH changes. We therefore consulted literature for critical Fe concentration levels, which cause interferences (Weishaar et al 2003; Xiao et al. 2013 and Poulin et al. 2014). We concluded that in our case Fe interference can be neglected as we measured maximum concentrations of 500 µg $L^{-1}$ Fe in our water samples. This results in a $Fe_{mg}/C_{mg}$ ratio of about 0.01, which is very low. Additionally, samples were measured in 6 or 8 fold dilutions resulting in maximum absolute concentrations during measurements of about 60 or 80 µg $L^{-1}$. Possible biases from pH were assessed by continuous pH measurements, which resulted in a constant pH of about 3.8 over the whole sampling period.

*# The fluorescence index (FI) differentiates between vascular plant derived (FI 1.3 – 1.4) and microbially derived (1.7-2.0) DOM. FI values >1.7 were interpreted as microbial DOM (p. 5 lines 15-20). Can the authors exclude that non-vascular plants (mosses) contributed to DOM generation? Later in the manuscript a contribution of Sphagnum is specifically discussed with respect to SUVA but not to FI (p. 9 lines 15-31).*

Response: We are glad that the referee pointed out this apparent limitation of this approach, which also caught our attention before. We do not exclude non-vascular plants, indeed we assume that bog derived DOM does also origin from prevailing Sphagnum vegetation. This fact should not modify assumptions that are made by the fluorescence index. The study of Wickland et al. (2007) published FI values of leachates of plant material. The FI of Sphagnum were around 1.1-1.4, which is similar to vascular plants signatures. We will rephrase the sentence to improve clarity here.

*# In the results and also in the discussion sections, fluorescence indices and absorption values were often reported to be higher or lower when sites or situations were compared. However, I missed statistics providing a little more confidence if these differences are significant. I*

*recommend including a table/figure, summarizing the main results as well as the levels of significance. This table/figure along with a hypothesis can be used to guide the reader through the results and discussion. I found it less convenient to work through the detailed description of the results. The figures 2, 3, 5, 6, 7 and 8 look more or less similar, which makes it not easy to keep the most important results in mind.*

Response: We agree that the current presentation was not easy to follow. We thus extend our study to clear statistical evaluations in the revised version. We will prepare descriptive statistic tables (planned as table 1 in the revised manuscript), which include mean, median, standard deviation and minimum and maximum values of the concentrations and indices. We will consider providing box plots to condense information and acknowledge the currently unclear presentation. Differences have now been tested with appropriate statistic tests like Mann-Whitney or Kruskal-Wallis tests and results are ready to be included in a revised version.

*# Further comments:*

*# The title: Please refer more directly to the outcome of the study*

Response: We will substantiate our title.

*# p.1 lines 22-24: The concluding sentence of the abstract "Our study demonstrated that DOM export dynamics are not only a passive mixing of different hydrological sources, but…" is unclear. Now the reader expects a statement why it is not a passive mixing or how the process can be characterized instead. However, it is only concluded that "…assessing DOM quality can greatly improve our understanding…". Please include here the most relevant result that improved our understanding. The sources and the quality of DOM appeared to be highly variable within events depending on runoff generation. What are the consequences for sampling/monitoring programs?*

Response: We will revise the abstract also in view of the comments of referee #1. We will refer stronger to the outcome of this study, which is the mentioned high variability of DOM quality. This implies that monitoring programs have to consider not only changes due to seasonality, but need to cover different hydrologic conditions as well. Furthermore, the study shows that bog DOM quality is less susceptible to changes in hydrologic conditions than the peaty riparian zones. Additionally, the used spectrofluorometric indices proofed to be a useful tool to track DOM origin and dynamics in this headwater stream.

*# p. 2 line 28: "aromatic or humic", is there a difference?*

Response: This is a tricky question. Well for our understanding the term "aromatic" refers to the specific ring structure in an organic molecule, while humic or humic substances rather describes a large fraction of DOM mainly defined as complex, high molecular weight, chromophoric molecules derived from decomposition of plant and animal residues. These substances comprise an aromatic fraction. We used those terms in account with the general usage in literature, especially in literature reporting fluorescence results (e.g. Inamdar et al. 2012).

*# p. 2 line 33: groundwater DOM is of smaller size and mostly of microbial origin, please include a reference*

Response: We will include a reference (e.g. Inamdar et al. 2012, Singh et al. 2012).

*# p. 4 line 29: The near UV includes light from 300 – 400 nm, 254 nm is in the middle UV range*

Response: The referee is right and we will change this.

*# p. 7 line 9: Figure 5 instead of Figure 6?*

Response: The referee is right. We will change this.

*# p. 8 lines 10 and 16-17: I had problems to relate these statements to the figures*

Response: We will illustrate that statement with box plot figures and tests of significant differences (Mann-Whitney or Kruskal-Wallis test).

*# p. 9 lines 6-9: A confirmation of the suitability of fluorometric indices appears difficult without independent methods (e.g. isotopes, mass spectrometry). It is problematic to conclude that an increase in aromaticity is caused by an increase in apparent molecular size if the latter is not measured.*

Response: Regarding the first part of this comment we agree with the referee that this statement might be misleading. This statement was not intended to refer to the suitability of fluorometric indices to represent specific DOM fractions or molecules. We want to express that the spectrofluorometric approach seems an appropriate tool to distinguish relevant hydrological compartments based on DOM quality. We will rephrase that.

Regarding the second part we conclude this from a negative correlation of the SUVA index for aromaticity and $S_R$ which is an index for molecular size as confirmed in available studies (see Helms et al. 2008). This is why we explicitly stated "apparent" molecular size to state that this is a molecular size derived from $S_R$. Maybe this was unclear. We will strengthen this assumption by a significant statistical correlation in the revised manuscript and will rephrase this conclusion in order of the method limitations.

*# p. 14 line 1: "The export of labile, protein-like DOM was specific…" I suggest being more cautious with characteristics of DOC that have not measured directly (e.g. "labile"). See also P. 9 line 5 "…specific strong microbial DOM signature…".*

Response: While in general the suitability of comparable fluorometric indices has been confirmed in existing studies (see e.g. Fellman et al. 2008), we agree that the method has inherent limitations and we will adjust these statements accordingly.

*# In Figures 1, 3, 5 and 6, the individual symbols in highlighted boxes (events) were difficult to distinguish.*

Response: We will consider this limitation in our revision of the figures. We will rework our figures to improve clarity.

References:

Fellman, J.B., D'Amore, D. V., Hood, E. and Boone, R. D.: Fluorescence characteristics and biodegradability of dissolved organic matter in forest and wetland soils from coastal temperate watersheds in southeast Alaska, Biogeochemistry, 88, 169-184, doi:10.1007/s10533-008-9203-x, 2008

Graeber, D., Gelbrecht, J., Pusch, M. T., Anlanger, C., and Schiller, D. von: Agriculture has changed the amount and composition of dissolved organic matter in Central European headwater streams, The Science of the total environment, 438, 435–446, doi:10.1016/j.scitotenv.2012.08.087, 2012.

Helms, J. R., Stubbins, A., Ritchie, J. D., Minor, E. C., Kieber, D. J., and Mopper, K.: Absorption spectral slopes and slope ratios as indicators of molecular weight, source, and photobleaching of chromophoric dissolved organic matter, Limnol Oceanogr, 53, 955–969, doi:10.4319/lo.2008.53.3.0955, 2008.

Hudson, N., Baker, A., Reynolds, D. M., Carliell-Marquet, C., and Ward, D.: Changes in freshwater organic matter fluorescence intensity with freezing/thawing and dehydration/rehydration, J. Geophys. Res., 114, doi:10.1029/2008JG000915, 2009.

Inamdar, S., Finger, N., Singh, S., Mitchell, M., Levia, D., Bais, H., Scott, D., and McHale, P.: Dissolved organic matter (DOM) concentration and quality in a forested mid-Atlantic watershed, USA, Biogeochemistry, 108, 55–76, 2012.

Poulin, B. A., Ryan, J. N., and Aiken, G. R.: Effects of Iron on Optical Properties of Dissolved Organic Matter, Environ Sci Technol, 48, 10098–10106, doi:10.1021/es502670r, 2014.

Santos, P. S. M., Otero, M., Santos, E. B. H., and Duarte, A. C.: Molecular fluorescence analysis of rainwater: effects of sample preservation, Talanta, 82, 1616–1621, doi:10.1016/j.talanta.2010.07.048, 2010.

Singh, S., Inamdar, S., Mitchell, M., and McHale, P.: Seasonal pattern of dissolved organic matter (DOM) in watershed sources: influence of hydrologic flow paths and autumn leaf fall, Biogeochemistry, 118, 321–337, doi:10.1007/s10533-013-9934-1, 2014

Weishaar, J. L., Aiken, G. R., Bergamaschi, B. A., Fram, M. S., Fujii, R., and Mopper, K.: Evaluation of Specific Ultraviolet Absorbance as an Indicator of the Chemical Composition and Reactivity of Dissolved Organic Carbon, Environ Sci Technol, 37, 4702–4708, doi:10.1021/es030360x, 2003.

Wickland, K. P., Neff, J. C., and Aiken, G. R.: Dissolved Organic Carbon in Alaskan Boreal Forest: Sources, Chemical Characteristics, and Biodegradability, Ecosystems, 10, 1323–1340, doi:10.1007/s10021-007-9101-4, 2007.

Xiao, Y.-H., Sara-Aho, T., Hartikainen, H., and Vähätalo, A. V.: Contribution of ferric iron to light absorption by chromophoric dissolved organic matter, Limnol Oceanogr-Meth, 653–662, 2013.

---

## Author Response (AR1)

**Author's Response to Referee's**

We would like to thank the two Referees for their helpful and constructive comments, which greatly improved the manuscript. Our detailed responses to the comments are listed below. For Author's changes see the attached revised manuscript with mark ups (track changes in MS Word) below. Page and line numbers of changes in the attached manuscript are given in brackets.

**Response to Referee #1**

*# The objectives could be improved by a clearer focus and formulation of hypotheses or research questions. The major drawback is the only fair scientific and presentation quality that needs to be improved. My major criticism is the simpified verbal description of the results that lacks quantitative assessments of concentrations/ component levels, seasonality and event behavior. Also, a quantitative statistical analysis on major governing factors such as temperature is needed.*

Response: In our revised manuscript we reworked the last section of our introduction to clarify our objectives and research question (p. 3 line 30 – p. 4 line 16). Additionally, we added statistical analysis (descriptive, as well as correlations and tests of significant differences) to our data to strengthen our statements (Table 1 p. 22, Figure 3 and 7 p. 27 and 33). These issues will be addressed in further detail in the specific comments of this response (see below).

*# Specific comments and technical corrections*

*# Abstract: For me, it reads a bit to technical and focussed on the results. I would like to see terms and variables like "C2%" to be reduced in favor of a stronger focus on discussion and implications (expanding the messages now in the last sentence).*

Response: As suggested by the referee we adjusted the focus of the abstract rather to discussion and conclusions and shortened the result section. We revised the abstract also in view of the comments of referee #2. We included major outcomes of the study like the high variability of DOM quality due to hydrologic conditions, which makes it necessary to cover discharge events during monitoring studies. We outline the differences between both studied catchments (bog/forested riparian zone) and shortly discuss the suitability of spectrofluorometric indices to track DOM origin and dynamics in this headwater stream. (p. 1 line 19 - p. 2 line 3)

*# Introduction: P2Line5: The first sentence need a reference.*

Response: We added a reference to that statement (Mulholland 2003). (p. 2 line 13)

*# P2Line 7f: This also needs a reference. Check also for this paper: Ledesma JLJ, Grabs T, Bishop KH, Schiff SL, Kohler SJ (2015) Potential for long-term transfer of dissolved organic carbon from riparian zones to streams in boreal catchments. Global Change Biology, 21, 2963-2979*

Response: We included references in the revised version of the manuscript (Fellman et al. 2009, Perdrial et al. 2014 and Wallin et al. 2015: p. 2 line 16-17). We refer to the suggested literature at another point in the introduction (p. 2 line 19).

*# P2L34: Change citation parenthesis.*

Response: We changed that in the revised manuscript not only at this particular point (p. 3 line 12), but we thoroughly read through the text and found several more parenthesis errors in the manuscript.

*# P3Line19: Better define what you mean with changes by adding space and time! Do you mean surface water DOM at the catchment outlet or variability within the study site – do you mean seasonal changes, daily fluctuations...: Something like "spatiotemporal dynamic of DOM quality over the course of one year" ...*

Response: We rephrased that sentence to avoid misunderstanding. The study aim described here is to elucidate different spatiotemporal dynamics in DOM quality over a year in a headwater stream, as the reviewer suggested. Those dynamics are supposed to be controlled by the bog and forested peaty riparian zone catchment. (p. 3 line 30-33)

*P3Line23: Two times starting a sentence with "on the other hand"*

Response: The section including this phrase has been reworked (p. 3 line 30 – p. 4 line 30).

*# P3Line31: This last section is not well integrated with the objectives in the section before: I suggest restructuring the objectives a bit starting from P3Line23; there is more or less everything there: What is the general aim? What are specific questions you want to answer/ Hypotheses to follow? Where do you perform this research (references on previous studies) and why there? What are the methods you want to apply? What are discussion/ conclusion and implications aiming at?*

Response: We agree with the referee and revised this section of the manuscript in view of the mentioned deficiencies and comments of referee #2. We clarify our objectives/hypothesis in view of these points: I) To test if spectrofluorometric indices can be used to track origin and dynamics of DOM. II) That DOM quality is highly variable in a headwater stream depending on hydrologic conditions and season. We expect short-term DOM quality changes due to high discharge events, which cause changes in hydrologic connectivity of different DOM pools and possible leachate effects of labile DOM during strong rain events. This short-term pattern is expected to be overlain by seasonal DOM changes due to changes of DOM production and consumption over the year. III) We expect general differences in DOM quality between the bog and forested riparian zone catchment. The riparian zone is characterized by accelerated water level fluctuations and a nutrient-richer vegetation than at the bog site, which leads to the hypothesis that DOM quality is strongly affected by changes in hydrologic conditions and is more labile than at the bog site. We point out the suitability of this particular catchment as the stream origins within the bog and enables us to retrieve an exclusively bog derived DOM signal within this headwater stream. We also refer to our previous study at this catchment and the applied spectrofluorometric methods, which are $SUVA_{254}$, $S_R$, FI, HIX and PARAFAC modeling of excitation emission matrices. (p. 3 line 30– p. 4 line 30)

*# Materials and methods: P4Line1-14: I am not convinced of the study site description in combination with Fig. 1: I suggest to show a soil map or something comparable to better see the position of the bog and the differences between the two sampling locations.*

Response: The provided Figure 1 is actually a soil map. We admit that the figure might not sufficiently highlight the information, which is important to this study. To make the figure more comprehensible we extended the figure caption and modified the map by clarifying the declaration of soil type distributions and sampling spots. (Figure 1, p. 23)

*# P4Line15: "Discharge sampling" sounds strange – just water sampling?*

Response: We agree with the referee and rephrase that expression. (p. 5 line 15)

*# P4L15ff: Please state the number of different types of samples and the temporal resolution of the discharge measurements..*

Response: We include this information now. In short, we took 30 grab samples, 44 water samples by the automated water sampler and 191 samples during high frequency storm event sampling. Discharge was recorded in a 10 minute time interval. (p. 5 line 16 – 23)

*# P4Line27: Unit of absorption?*

Response: The missing unit of absorption is m$^{-1}$. We corrected this. (p. 5 line 30)

**P4Line29f: Superscript in units!**

Response: We edited this mistake. (p. 6 line 1)

**Results: Fig. 2: What do you mean by "trends during rain events"? I suppose the concentration trends?**

Response: We agree that this is a poor expression. We rephraseed it to "trend of concentration" or "trend of quality indices", respectively. (Figure 2, p. 24 line 4)

**P5Line29f: Please put that statement in numbers (e.g. CV or standard deviation of seasonal concentrations and storm event concentrations).**

Response: We picked up the referees comment and strengthened our statements with a table (Table 1, page 22) of descriptive statistics including mean, median, standard deviation and minimum and maximum values for the annual record. For the event data set we present the data more comprehensively in boxplots figures (Figure 3 and 7 p. 25 and 31). Additionally, we provide a table with this event data in the supplement for better comparability of numbers (see supplement, S1). We now refer to specific numbers of the tables in the text if needed (like standard deviations). (p. 7 line 9ff)

**P5Line30f: You mean the highest recorded concentrations during the entire study period? This is not clear from this sentence.**

Response: The referee is right, the statement refers to the entire study period. We rephrased that sentence for clarification. (p. 7 line 12-13)

**P6Line1: Again – put that into numbers (e.g. mean or median concentrations)**

Response: As described to a previous comment we included a new table, which will provide the requested numbers and added necessary information in the text. (Table 1, p. 22 and p. 7 line 16ff)

**P6Line23ff: I struggle with the interpretation of the components as a "too early" discussion. Maybe a introduction sentence on that?**

Response: We included the interpretation of the components in the results after careful consideration and with regard to existing literature to facilitate reading and help the reader understand the common use of these indices. Therefore, we do not consider this as a real "discussion" here. If we include this information only in the discussion section, the result section describing characteristics of different rain events would be more difficult to follow, e.g. as the description of components would remain very technical and abstract. Furthermore the focus of the discussion would be less clear, if we introduce the common interpretation of components only there. From our point of view this interpretation is thus rather a description of the modelled components, in terms of 'results', backed up by literature data (which is admittedly unusual for a normal result section). Such description is found also in the results or method sections in other studies. Nevertheless, to address this unease of an early discussion we included an introducing sentence now. (p. 8 line 9-10)

*# P7Line19ff: Again, I have problems with this type of interpretation in the result section. This more belongs to the discussion chapter.*

Response: We shifted the interpreting statements of this sentences (now p. 9 line 7-9) to the discussion section. We now limit the description solely to the components and their contributions (referring to studies identifying similar components) without assigning a source of these components here, but putting this source assignment into the discussion. (p. 12 line 11-13)

*# In general I am not convinced of the discussion of seasonal trends: For a seasonality assessment I would expect a more in depth evaluation of seasonal min and max and potential controls: e.g. Is the seasonality in line with water temperature (which is likely close to soil water temperature), air temperature, light intensity... So, better quantify and describe the seasonality!*

Response: We thank the referee for this constructive comment. We agree that statistical indications are helpful to underline our statements in regard to seasonality. We include a description of minimum and maximum values now (Table 1 p. 22 and Fig. 3 and 7 p. 25 and 31). Unfortunately, we did not measure soil temperature and only occasionally water temperature, which limits options to further evaluate seasonality effects as suggested. However, we chose to use air temperature as a proxy for seasonality and included results from Spearman rank correlations in the manuscript (p. 7 line 14-16 and p. 11 line 32 – p. 12 line 1).

*# Discussion: P9Line 6f: This type of statement belongs to the end of discussion/ conclusions!*

Response: We agree with the referee and moved this point to the conclusions in the revised manuscript. (p. 11 line 2-3 and p. 16 line 21-22)

**Response to Referee #2**

**Specific comments:**

*# The study is intended to elucidate the DOM quality and dynamics in two contrasting catchments. The authors expect "water level, temperature and precipitation to be the main controls on stream DOM…". I would acknowledge if the authors provide a more specifically defined working question/hypothesis along which the results and discussion can be organized. How can the DOM of both catchments differ?*

Response: Also in line with referee #1 we rephrased our objectives in the introduction to clarify our hypothesis and provide a better readability (p. 3 line 30 – p. 4 line 30). Main objectives/hypotheses were: I) To test if spectrofluorometric indices can be used to track origin and dynamics of DOM. II) That DOM quality is highly variable in a headwater stream depending on hydrologic conditions and season. We expect short-term DOM quality changes due to high discharge events, which cause changes in hydrologic connectivity of different DOM pools and possible leachate effects of labile DOM during strong rain events. This short-term pattern is expected to be overlain by seasonal DOM changes due to changes of DOM production and consumption over the year. III) We expect general differences in DOM quality between the bog and forested riparian zone catchment. The riparian zone is characterized by accelerated water level fluctuations and a nutrient-richer vegetation than at the bog site, which leads to the hypothesis that DOM quality is strongly affected by changes in hydrologic conditions and is more labile than at the bog site. Additionally, we reviewed our result and discussion section in term of hypothesis consistency.

*# Samples were taken by an automated sampler in a six day interval and the filtered samples were stored at 4°C in the dark. How long were the samples stored until analysis? Although filtered, the samples were probably not sterile. Bacteria can degrade organic carbon even at low temperatures. Have the authors tested if the DOC composition remained unchanged?*

Response: The storage time of our samples depended upon the type of measurement. DOC concentration and UV-Vis analytics were conducted at the TU Braunschweig. We could therefore measure samples more or less immediately (within a week). Immediate fluorescence measurements were impossible, as those were conducted at the University of Münster, which necessitated measurement campaigns in about three months intervals. To address the problem of conservation we tested UV-VIS changes, which were inconspicuous over that time-period. Furthermore, due to the longtime experience of the ecohydrology group in Münster in fluorescence measurements we came to the conclusion that we produce less analytic artefacts by storing the samples at low temperatures over that time period than freeze them for conservation. We also refer to following literature: Santos et al. 2010, Hudson et al. 2009, Graeber et al. 2012. There it also is recommended that samples should be frozen, if storage time is longer than one year.

*# UV absorption characteristics were used as indicators of DOM composition (absorbance at 254 nm, spectral slope ratio). Besides DOM, however, dissolved iron exhibits significant UV absorption. The dynamics of iron is also related to hydrology, i.e. flood events can be associated with high iron concentrations. Changes in UV absorption may reveal changes in DOM quality but at the same time they can reflect different contributions of dissolved iron. How the authors think about that?*

Response: This is an important point. We are aware of biases of UV-VIS and fluorescence DOM characterization by Fe concentration or pH changes. We therefore consulted literature for critical Fe concentration levels, which cause interferences (Weishaar et al 2003; Xiao et al. 2013 and Poulin et al. 2014). We concluded that in our case Fe interference can be neglected as we measured maximum concentrations of 500 µg $L^{-1}$ Fe in our water samples. This results in a $Fe_{mg}/C_{mg}$ ratio of about 0.01, which is very low. Additionally, samples were measured in 6 or 8 fold dilutions resulting in maximum absolute concentrations during measurements of about 60 or 80 µg $L^{-1}$. Possible biases from pH were assessed by continuous pH measurements, which resulted in a constant pH of about 3.8 over the whole sampling period.

*# The fluorescence index (FI) differentiates between vascular plant derived (FI 1.3 – 1.4) and microbially derived (1.7-2.0) DOM. FI values >1.7 were interpreted as microbial DOM (p. 5 lines 15-20). Can the authors exclude that non-vascular plants (mosses) contributed to DOM generation? Later in the manuscript a contribution of Sphagnum is specifically discussed with respect to SUVA but not to FI (p. 9 lines 15-31).*

Response: We are glad that the referee pointed out this apparent limitation of this approach, which also caught our attention before. We do not exclude non-vascular plants, indeed we assume that bog derived DOM does also origin from prevailing Sphagnum vegetation. This fact should not modify assumptions that are made by the fluorescence index. The study of Wickland et al. (2007) published FI values of leachates of plant material. The FI of Sphagnum were around 1.1-1.4, which is similar to vascular plants signatures. We will rephrase the sentence to improve clarity here. (p. 6 line 17)

*# In the results and also in the discussion sections, fluorescence indices and absorption values were often reported to be higher or lower when sites or situations were compared. However, I missed statistics providing a little more confidence if these differences are significant. I recommend including a table/figure, summarizing the main results as well as the levels of significance. This table/figure along with a hypothesis can be used to guide the reader through the results and discussion. I found it less convenient to work through the detailed description of the results. The figures 2, 3, 5, 6, 7 and 8 look more or less similar, which makes it not easy to keep the most important results in mind.*

Response: We agree that the former presentation was not easy to follow. We thus extended our study to clear statistical evaluations in the revised version. We prepared descriptive statistic tables (Table 1 p. 22 and S1 in the supplement), which include mean, median, standard deviation and minimum and maximum values of the concentrations and indices. We provide box plots to condense information and acknowledge the formerly unclear presentation (Figure 3 and 7 p. 25 and 31). Differences have now been tested with appropriate statistic tests like Mann-Whitney or Kruskal-Wallis tests and results are included in the boxplots (p. 6 line 23-30 and Figure 3 and 7).

*# Further comments:*

*# The title: Please refer more directly to the outcome of the study*

Response: We substantiated our title and changed it to "Changes in dissolved organic matter quality in a peatland and forest headwater stream as a function of seasonality and hydrologic conditions" (P.1 line 2-4).

*# p.1 lines 22-24: The concluding sentence of the abstract "Our study demonstrated that DOM export dynamics are not only a passive mixing of different hydrological sources, but…" is unclear. Now the reader expects a statement why it is not a passive mixing or how the process can be characterized instead. However, it is only concluded that "…assessing DOM quality can greatly improve our understanding…". Please include here the most relevant result that improved our understanding. The sources and the quality of DOM appeared to be highly variable within events depending on runoff generation. What are the consequences for sampling/monitoring programs?*

Response: We revised the abstract also in view of the comments of referee #1. We now refer stronger to the outcome of this study, which is the mentioned high variability of DOM quality. This implies that monitoring programs have to consider not only changes due to seasonality, but need to cover different hydrologic conditions as well. Furthermore, the study shows that bog DOM quality is less susceptible to changes in hydrologic conditions than the peaty riparian zones. Additionally, the used spectrofluorometric indices proofed to be a useful tool to track DOM origin and dynamics in this headwater stream. (p. 1 line 18 – p. 2 line3)

*# p. 2 line 28: "aromatic or humic", is there a difference?*

Response: This is a tricky question. Well for our understanding the term "aromatic" refers to the specific ring structure in an organic molecule, while humic or humic substances rather describes a large fraction of DOM mainly defined as complex, high molecular weight, chromophoric molecules derived from decomposition of plant and animal residues. These

substances comprise an aromatic fraction. We used those terms in account with the general usage in literature, especially in literature reporting fluorescence results (e.g. Inamdar et al. 2012).

*# p. 2 line 33: groundwater DOM is of smaller size and mostly of microbial origin, please include a reference*

Response: We now refer to the references Inamdar et al. 2012 and Singh et al. 2012. (p. 3 line 10)

*# p. 4 line 29: The near UV includes light from 300 – 400 nm, 254 nm is in the middle UV range*

Response: The referee is right and we changed this. (p. 5 line 31)

*# p. 7 line 9: Figure 5 instead of Figure 6?*

Response: The referee is right. We changed this reference to Figure 4 now, due to changes in the Figures. (p. 8 line 28)

*# p. 8 lines 10 and 16-17: I had problems to relate these statements to the figures*

Response: We deleted the first statement ("$S_R$ exhibited generally higher values during both spring events indicating smaller DOM molecular size at the forested site") (p. 10 line 2-3), rephrased the second one (now: "Contribution of protein-like component C5% exhibited elevated values during the second spring event with wet preconditions at the bog site (Fig. 6, 7)) and included a new reference to the boxplot figures (p. 10 line 8-9).

*# p. 9 lines 6-9: A confirmation of the suitability of fluorometric indices appears difficult without independent methods (e.g. isotopes, mass spectrometry). It is problematic to conclude that an increase in aromaticity is caused by an increase in apparent molecular size if the latter is not measured.*

Response: Regarding the first part of this comment we agree with the referee that this statement might be misleading. This statement was not intended to refer to the suitability of fluorometric indices to represent specific DOM fractions or molecules. We want to express that the spectrofluorometric approach seems an appropriate tool to distinguish relevant hydrological compartments based on DOM quality. We will rephrase that. (p. 11 line 1-2)

Regarding the second part we conclude this from a negative correlation of the SUVA index for aromaticity and $S_R$ which is an index for molecular size as confirmed in available studies (see Helms et al. 2008). This is why we explicitly stated "apparent" molecular size to state that this is a molecular size derived from $S_R$. Maybe this was unclear. We strengthened this assumption by a significant statistical correlation in the revised manuscript (p.7 line 28-30) and rephrased this conclusion in order of the method limitations. (p. 11 line 2-6)

*# p. 14 line 1: "The export of labile, protein-like DOM was specific…" I suggest being more cautious with characteristics of DOC that have not measured directly (e.g. "labile"). See also P. 9 line 5 "…specific strong microbial DOM signature…".*

Response: While in general the suitability of comparable fluorometric indices has been confirmed in existing studies (see e.g. Fellman et al. 2008), we agree that the method has inherent limitations and we adjusted these statements accordingly. (p. 16 line 11-12 and p. 16 line 14-17)

*# In Figures 1, 3, 5 and 6, the individual symbols in highlighted boxes (events) were difficult to distinguish.*

Response: We considered this limitation in our revision of the figures. We changed the data point marks and display some results in boxplots now. (Fig. 2 – 7, p. 24 – 33)

[revised manuscript text omitted]

---

## Referee Report (RR1)

hess-2016-377

Changes in dissolved organic matter quality in a peatland and forest headwater stream as a function of seasonality and hydrologic conditions by Broder et al.

Review:

P1L20: „was also particularly exported" is a strange expression – just „also" or „particularly" is sufficient (depending on what you want to say)

Figure 1: I am still a bit puzzled here: Why are both observations points within one catchment? It would make much more sense to distuinguish catchments of each point, land use and soils. The dashed line is distracting.

P17L6: „SD" means standard deviation? Why do you switch between precision here? Sometimes you use no, sometimes one digit for the concentration. In table 1 its 2! Please adjust that to a meaningful precision.

P6L18: use „larger" instead of „greater"

P7L2: type „but showed"

---

## Author Response (AR3)

Dear Editor,

I am very happy that only minor iterations are necessary! Thank you very much for the thorough editing of our manuscript. I added a description of the arrows in the caption of Figure 6. However, I have trouble with the formatting issue of Table 1. I changed a small formatting error were a line was drawn one cell too long. But I guess you mean the apparent small lines, which indicate cell boundaries?! I was not able to delete those, they always appear when converting to a pdf. However, when you print the document or zoom in, those small lines disappear, so I think it is just a display problem?

Best regards,

Tanja Broder